



# Modeling tsunami initial conditions due to rapid coseismic seafloor displacement: efficient numerical integration and a tool to build unit source databases

Alice Abbate[1,2], José M. González Vida[3], Manuel J. Castro Díaz[3], Fabrizio Romano[1], Hafize Başak Bayraktar[1], Andrey Babeyko[4], and Stefano Lorito[1]

[1]Istituto Nazionale di Geofisica e Vulcanologia (INGV), Rome, Italy
[2]Department of Mathematics, Informatics and Geosciences (MIGe), University of Trieste, Trieste, Italy
[3]Department of Applied Mathematics, EDANYA group, University of Malaga, Malaga, Spain
[4]GFZ German Research Center for Geosciences, Potsdam, Germany

**Correspondence:** Alice Abbate (alice.abbate@ingv.it)

**Abstract.** The initial condition for the simulation of a seismically-induced tsunami for a rapid, assumed instantaneous, vertical seafloor displacement is given by the Kajiura low-pass filter integral. This work proposes a new efficient and accurate approach for its numerical evaluation, valid when the sea floor displacement is discretized as a set of rectangular contributions. We compare several truncated quadrature formulae, selecting the optimal one. We verify that we can satisfactorily approximate the

initial sea level perturbation as a linear combination of those induced by the elementary sea floor displacements. The methodology is tested on the tsunamigenic Kuril earthquake doublet - a megathrust and an outer-rise - occurred in the Central Kuril Islands in late 2006 and early 2007. We also confirm the importance of the horizontal contribution to the tsunami generation and we consider a simple model of the inelastic deformation of the wedge, on a realistic bathymetry. The proposed approach results accurate and fast enough to be considered relevant for practical applications, and a tool is provided to create tsunami

unit source databases for a given region of interest.

## 1 Introduction

The generation of a seismotectonic tsunami occurs when the equilibrium of the water column is perturbed by the seafloor deformation induced by an earthquake. Seismic and oceanic acoustic waves are radiated from the source and contribute to the total wave field, as evidenced by the records of bottom pressure sensors (Abrahams et al., 2023). Models which solve

for the full bidirectional coupling between the seafloor and the ocean have been developed (e.g. Maeda and Furumura 2013; Lotto and Dunham 2015). An approximate two-stage procedure has also been proposed, which solves the tsunami excitation as the result of a time-dependent seafloor displacement in a compressible ocean, and then propagates the wave train through an incompressible one (Saito et al., 2019). Seismic and oceanic acoustic waves are driven by the elasticity of the medium and travel, on average, two orders of magnitude faster than tsunamis, which are moved by gravity (Saito, 2019). This fact leaves

room to decouple seismic contributions from tsunami waves, in the sense that it is reasonable to assume this time window as too short to convert seismic energy into fluid potential energy (Pedersen, 2001; Nosov and Kolesov, 2007). Hence, the





excitation of a tsunami can be described with the linear potential theory for an incompressible and irrotational fluid, perturbed by a bottom dislocation significantly smaller than the sea depth (Saito, 2013, 2019). In this framework, the velocity of fluid displacement is expressed in terms of a scalar potential satisfying the time-dependent Laplace problem (Lamb, 1945; Stoker,

1958; Landau and Lifshitz, 1987). Analytic solutions for the sea surface height distribution have been derived both in time and Fourier domain, sometimes benchmarked against laboratory experiments (Hammack, 1973; Comer, 1984; Dutykh et al., 2006; Dutykh and Dias, 2007; Saito, 2013; Levin and Nosov, 2009). A numerical solution to the full Laplacian problem has also been proposed (Nosov and Kolesov, 2009; Rabinovich et al., 2008). It has been discussed that these solutions may be necessary for earthquakes characterized by steep dip angle or prolonged source duration (Kajiura, 1970; Kervella and Dutykh, 2007; Saito

and Furumura, 2009; Madden et al., 2020).

   However, if the events, like ordinary megathrust earthquakes are often assumed to do, take place within a sufficiently brief time frame (compared to the tsunami propagation scale), the seafloor deformation can be treated as instantaneous (Abrahams et al., 2023; Nosov and Kolesov, 2011). For relatively long-wave displacements, the initial condition for modeling tsunami propagation is then typically obtained by copying the static permanent vertical coseismic deformation of the seafloor at the

free surface. The contribution of the horizontal component to the coseismic deformation can also be important in the presence of steep slopes in the bathymetry (Iwasaki, 1982; Tanioka and Satake, 1996), or in shallow earthquakes resulting in an additional uplift in the accretionary prism (Seno, 2000; Tanioka and Seno, 2001). Some approaches impose a delta function as the bottom velocity (Levin and Nosov, 2009; Saito, 2017) or transfer to the sea-level the last frame of a time-dependent earthquake rupture simulation (Saito, 2019; Abrahams et al., 2023). The linear potential theory generally yields a sea surface perturba-

tion that does not coincide with the coseismic seafloor displacement (Ward, 2003; Kajiura, 1963; Saito, 2013). Kajiura (1963) demonstrated analytically that, in the hypothesis of an instantaneously displaced flat bathymetry, the sea surface perturbation can be formally expressed in terms of a Green's function. In this expression, the waves characterised by $kH >> 1$ are progressively more damped by a factor $\frac{1}{\cosh(kH)}$, where $k$ is the wavenumber and $H$ is the sea depth. These findings hold significance for numerical modeling of the linear generation process, highlighting the need to filter out high frequency spectral compo-

nents to prevent the introduction of non-physical artifacts. Filtering of the short wavelengths becomes crucial when modeling real events whose lateral rupture extent is comparable to the ocean depth or in cases of residual deformation characterized by small scale heterogeneities along the horizontal direction (Nosov and Kolesov, 2011); non-physical short waves would result in an overestimation of the initial condition and thus of the ensuing tsunami. A widely used method involves the application of the "Kajiura-type" filter. The initial condition for simulating tsunamis is thus obtained through Fourier integration of the

coseismic deformation, divided by the damping term $\frac{1}{\cosh(kH)}$. In this approach, $H$ represents the average depth within the region affected by coseismic deformation. The validity of such assumption is elucidated in the work of Abrahams et al. (2023). In Eq. (39), the authors examine a "transfer function" in the Fourier domain, which establish the response of the free surface (system output) when an arbitrary instantaneously moving ocean floor (system input) occurs. While the Kajiura filter can be applied to a displacement of virtually any shape, Davies and Griffin (2018) explain that the initial static condition resulting

from the instantaneous and simultaneous displacement of different subfaults can be obtained through a linear combination of elementary contributions, each represented by a Green's function. These unit contributions pertain to discrete portions of the





area of interest, taking into account variable sea-depth along the domain, which must be approximated as constant for each of the subfaults. This is a relatively cheap solution than the full implementation of the three dimensional Laplace problem whose degree of approximation with respect to a fully variable sea depth should be tested. On the other hand, Nosov and Kolesov
(2011) introduced a specific analytical solution to the 3D Laplace problem. This solution also assumes an instantaneously displaced flat seafloor within a rectangular region. Due to the fast decay of such solution, the free surface perturbation becomes negligible at the distance of $4H$. This allows the initial condition for the coseismic displacement to be approximated by linear combination of elementary contributions.

The authors developed a "Laplace smoothing algorithm" and discussed that this procedure is valid for a constant bathymetry,
but the approximation still holds reasonably if the bathymetry varies smoothly within a short distance ($\sim 4H$) from the source (a similar argument likely applies also for the combination of subfaults as proposed by Davies and Griffin, 2018). Nosov and Sementsov (2014) subsequently demonstrated its accuracy by comparing it with the case of variable bathymetry. However, even with these simplifying assumptions, the numerical integration of the model and the application of the complete algorithm get long execution times, both because of the extension of the integral support in the Fourier domain and of the desired resolution
in the spatial domain, which may lead from ten of hundreds to tens of millions of elementary initial conditions to be evaluated and superposed. The reliability of the solution and the computational time needed strongly depend on the quadrature method adopted to deal numerically with such a model. This reasoning leaves room to build from scratch an alternative implementation of the algorithm, which is the topic of this study.

The paper is structured a follows: first, for simplicity, we tackle the problem in one dimension (Section 2). We investigate the
convergence of the integral, defining the analytical error when truncating its domain. We then identify the optimal quadrature formula in terms of efficiency and accuracy. Moving to the two dimensional case, we describe a tool based on the idea of a pre-computed database of filtered unit initial conditions. Such conditions, functions of the local sea depth, can be linearly combined to obtain a discretization of any sea floor displacement (Section 3). Finally, to validate our approach, we test our algorithm on the Central Kuril earthquake doublet, a megathrust and an outer-rise event, occurred in late 2006 and early 2007 (Section
4), comparing our results with other studies addressing a similar problem (Nosov and Kolesov, 2011, 2009; Rabinovich et al., 2008).

## 2   Unit tsunamigenic source: numerical solution in one dimension

We consider a domain $D \subset \mathbf{R}$ of real numbers. $D$ is partitioned into $N_c$ sub-intervals $\{c_i\}_{i=1}^{N_c}$ of constant length $a \in \mathbf{R}$ and $x \in D$ is a point in this domain. We may think of the domain $D$ as a track in an ocean basin, whose points represent geographic
coordinates, in which both bathymetry and coseismic seafloor deformation are defined (Fig. 1).

Within each cell $c_i$, the sea surface height distribution (which we call also perturbation in the following) due to an instantaneous, uniform sea bottom displacement $\eta_0^i$, is represented by a box-car function obtained as the difference of two Heaviside functions as:




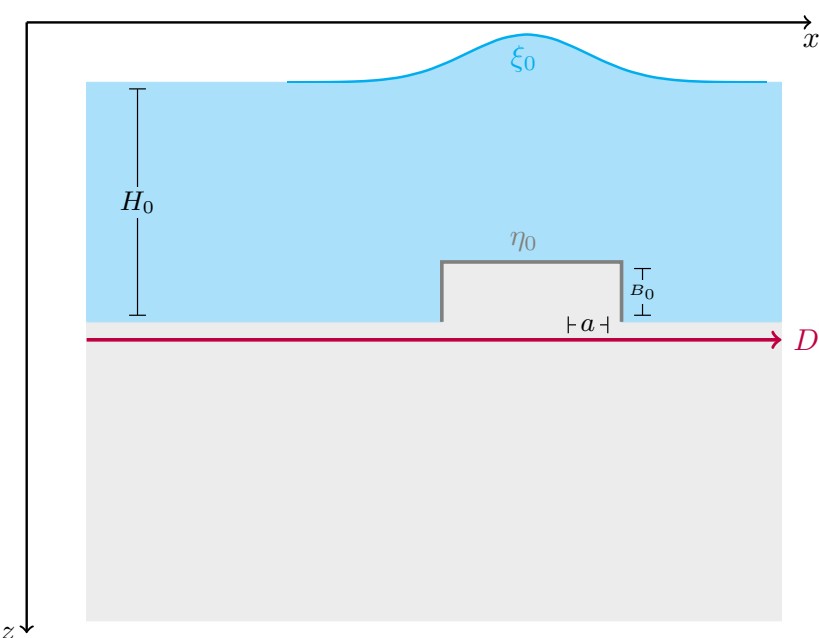

**Figure 1.** Schematic diagram of the problem in one dimension. The domain $D$ is partitioned into cells of equal length $a$. In each cell, $B_0$ is the amplitude of the coseismic deformation $\eta_0$, modeled as the difference of two Heaviside functions. In the figure, only an upward displacement in several adjacent cells with a constant water depth $H_0$ is sketched together with its effect $\xi_0$ at the sea surface. In reality, faulting would displace several cells either upward or downward in a more complex manner, and the water depth $H_0$ varies from place to place.

$$\eta_0^i = B_0^i\big[\theta(x+a) - \theta(x-a)\big] \tag{1}$$

The corresponding sea surface perturbation is given by (Nosov and Sementsov, 2014):

$$\xi_0^i(x) = \frac{2B_0^i}{\pi} \int\limits_0^\infty \frac{cos(mx)sin(ma)}{mcosh(mH_0^i)} dm \tag{2}$$

where $B_0^i$ is the amplitude of $\eta_0^i$ and $H_0^i$ the sea-depth in $c_i$. Equation (2) is valid for a flat bathymetry $H_0^i$. However, Nosov and Sementsov (2014) demonstrated its validity also for an arbitrary sloping bathymetry.

The variable of integration $m$ represents the spatial wavenumber and quantifies the number of oscillations of the integrand function in the domain of integration. The term $F(k, H_0) = \frac{1}{cosh(mH_0^i)}$ appearing in Eq. (2) is a "Kajiura-type" filter, which tends toward zero as $mH_0^i >> 1$, indicating that small wavelengths ($\lambda << H_0^i$) are effectively attenuated. The free surface



perturbation $\xi_0^i$ is also smooth, as it is derived analytically from the Laplacian problem. Each cell $c_i$ is associated to what we will call, from now on, the *Local Extended Domain* (LED):

$$l_e^i = \left( -4H_0^i - \frac{a}{2}, \; 4H_0^i + \frac{a}{2} \right) \tag{3}$$

whose extension depends on the sea depth. The initial condition given by Eq. (2) is solved numerically in every point $x_p \in l_e^i$. For all the points outside the LED (3), the free-surface perturbation vanishes asymptotically (Nosov and Kolesov, 2011).

The unit initial conditions $\xi_0^i$ generated by the bottom deformation within each segment $c_i$ must be later combined to obtain the final sea surface perturbation (the tsunami initial condition) over the total domain $D$. In this section, we examine a unit cell $c_i$ within the domain $D$, where the deformed seafloor (1) perturbs the free surface as in Eq. (2). For simplicity, we temporarily 105 exclude the superscript $i$ when considering only one cell.

To solve the integral numerically, we follow these steps:

1. Restrict the wavenumbers involved in the integration to a limited subset $[0, U]$, where $U$ is determined through tolerance tests for various parameterizations of the model (2);

2. Identify the optimal quadrature method by comparing different solutions in terms of accuracy and computational effi-
ciency.

More detailed informations are provided in the Supplementary Materials.

## 2.1 Corner wavenumber for truncation

We seek for upper-limiting the integration interval to a finite value of $U$ enabling us to solve the integration only for a reduced subset of wave numbers. Equation (2) can be restated as:

$$\xi_0(x) = \frac{2B_0}{\pi} \left( a\epsilon + O(\epsilon^3) + \int_\epsilon^U \frac{\cos(mx)\sin(ma)}{m\cosh(mH_0)} dm + o(e^{-\frac{UH_0}{2}}) \right) \tag{4}$$

This is relevant to save computational time and to determine which wavelengths should be filtered out when transferring the sea bottom deformation to the sea surface level. We consider the cell size $s \in \{15, 30, 60\}$, where the units are given in arc-seconds (hereafter referred to as arc-sec). This set of values is commonly adopted when modeling tsunamis. We then take a set of incremental discrete values for the local depth $d \in \{1\,\text{km}, 2\,\text{km}, \dots, 8\,\text{km}\}$. Each pair $(s, d)$ is associated to a LED
(3) and to a free surface height distribution $\widehat{\xi_0^{s,d}}$ (2). We employ a *Global Adaptive Quadrature* (Shampine, 2008), hereafter identified by the acronym GAQ, as the reference solution for each of the $3 \times 24$ combinations of parameters $s$ and $d$. For each depth value $d$, we then consider a range of possible upper limits $u_{j,d} = \frac{j}{d}$, where $j \in \{0.5, 1, 1.5, \dots, 5\}$ for the support of the integral as in Eq. (4). Each of the $3 \times 24 \times 10$ combinations of cell size $s$, water depth $d$ and integral UL (upper limit) $u_{j,d}$ is associated to a sea height distribution $\xi_0^{s,j,d}$ within a truncated support, according to Eq. (4), which is solved numerically


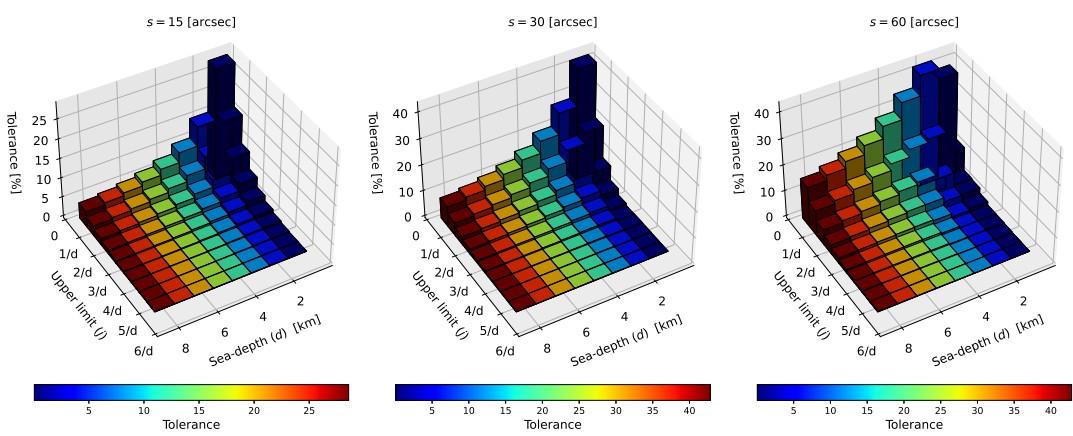

**Figure 2.** To identify the wavenumbers that play a substantial role in transferring seafloor deformation to the sea surface, we assess the tolerance by solving Eq. (4) with varying upper limits in the support of the integral. Additionally, we consider different model parameterizations, such as cell size and sea depth.

making use of GAQ as before. For each $j$, the results of the truncated integration are compared to the reference solutions in terms of Maximum Absolute Error (MAE):

$$e^{s,d,j} = max_x \left| \xi_0^{s,d,j}(x) - \widehat{\xi_0^{s,d}(x)} \right|$$

The upper limit for the support of the integral in Eq. (4) can be defined depending on the desired tolerance level for a specific cell size. It can be seen from Fig. 2 that, given $u_{j,d}$, tolerance generally increases when considering longer cells and shallower

water, with some exceptions (see for instance the third graph corresponding to a cell size of 60 arc-sec). However, for all the





cell sizes and given a depth value, the tolerance decreases if $u_{j,d}$ increases, as it can be expected from the theoretical error. For all the combination of parameters, the maximum tolerances are less than 50 % and approach zero for $u_{j,d} \geq \frac{3}{d}$. However, if the support of the integral contains few wavenumbers, the tails of its numerical solution may not be stable. To avoid this problem, we finally set the upper bound of the integral to be used in Eq. (4) as $U = \frac{5}{H_0}$, for all the possible values of $H_0$, which is aligned

with the convergence condition requiring $U > \frac{2}{H_0}$. The direct consequence is that all the wavelengths $\lambda < \frac{H_0}{5}$ will be filtered out in transferring seafloor deformation to the sea surface.

## 2.2   Optimal quadrature method for numerically solving the integral

Since both sine and cosine cannot be greater than one, Eq. (4) can be re-stated in a more convenient scaled version:

$$\xi_0(x_p) \simeq U \frac{2B_0}{\pi} \left( \frac{a\epsilon}{U} + \int_{m=\frac{\epsilon}{U}}^{1} \frac{cos(mUx_p)sin(mUa)}{mUcosh(mUH_0)} \right) dm \qquad (5)$$

for each point $x_p$ in the LED (3) of the cell. From Section (2.1), $U = \frac{5}{H_0}$ and we set $\epsilon = 10^{-9}$. It is important to emphasize that the integration domain, $\left[\frac{\epsilon}{U}, 1\right]$, does not align with the spatial domain (3). The former is associated with the wavenumber and expresses the physical wavelengths considered when modeling the sea surface after an instantaneous earthquake, $\left[\frac{H_0}{5}, 10^9\right]$. The latter represents the discretization of the seafloor displacement into cells of equal length $a$. Equation (5) is an approximation of the seabed deformation to the sea surface within a single cell, whose influence extends to all neighboring cells within a

distance of $|4H_0|$ from the center. It should be recalled that the approximation is valid when both the bathymetry and coseismic displacement vary slowly within such a radius. The integrand function in Eq. (5) exhibits significant oscillations, calling for the use of an adaptive composite formula for quadrature computation. The goal of a composite adaptive formula is to optimally partition the support intervals, with the algorithm dynamically selecting the number of sub-intervals. Numerical integration is then executed in each sub-interval of the support, and the final result is obtained by summing the contributions of the solution

within each sub-interval. In Eq. (5), the partitioning of the integral support should be determined based on the numerator of the integrand function, denoted as $g(mK, x, a) = cos(mKx)sin(mKa)$. This function represents the product of a cosine and a sine, oscillating at two distinct characteristic frequencies, depending on the point $x_p$ and the cell length $a$. Achieving this requires dividing the integral support into a number of points regulated by the maximum frequency of oscillation:

$$w_{max} = U max\left(\frac{x_p}{2\pi}, \frac{a}{2\pi}\right) \qquad (6)$$

Integration is a loop over the points $x_p$ in LED (3). At each iteration, the free surface deformation is found by solving Eq. (5) into a number

$$N_m = max\left[2w_{max}, N_s\right] \qquad (7)$$

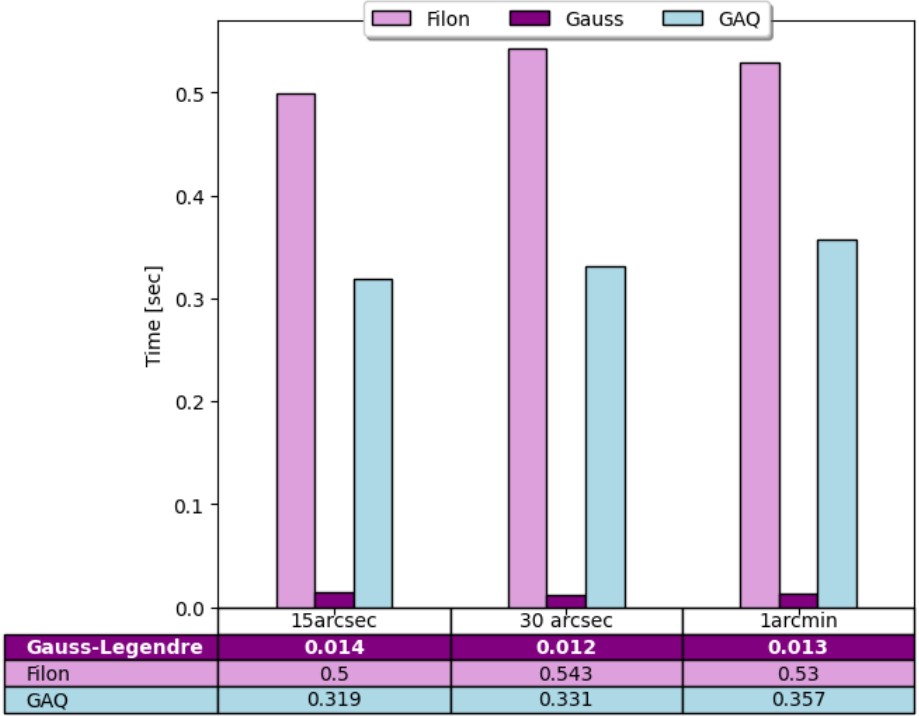

| | 15arcsec | 30 arcsec | 1arcmin |
|---|---|---|---|
| **Gauss-Legendre** | **0.014** | **0.012** | **0.013** |
| Filon | 0.5 | 0.543 | 0.53 |
| GAQ | 0.319 | 0.331 | 0.357 |

**Figure 3.** The efficiency of the different adaptive quadrature formulas (GLQ, FQ, GAQ) is illustrated in the histograms. For each cell size, the computation time, averaged across the eight depth values, is shown.

of partitions of the integral support, according to the Nyquist theorem. The variable $N_s$ is the minimum number of sub-supports needed to properly capture the sinusoidal cycles. This number is assessed by trial and errors over a large number of
different model parametrizations as $N_s = 20$. Two different quadrature methods are compared:

1. The Gauss-Legendre quadrature with three points (justified by the harmonic nature of the analytical solution to the problem (5), hereafter called GLQ;

2. The Filon-type quadrature, which is well known to be efficient in case of highly oscillating integrands (Filon, 1930; Iserles, 2004). We will refer to it as FQ.

The above-mentioned methodologies are adapted by following Eq. (6) to (7). All the details are presented in the Supplementary Materials. The deformed free-surface $\xi_0$ is found for three different cell sizes (15 arc-sec, 30 arc-sec, 60 arc-sec) and for eight depth values, ranging from 1 km to 8 km every 1 km. Results are checked against the reference solution (GAQ) as in Section (2.1). We compare the algorithms in terms of their efficiency (execution time) and accuracy. The efficiency is measured considering the average execution time of three runs. The accuracy is provided as Root Mean Squared Error (RMSE), averaged
over all the considered sea-depths.




We find that the RMSE between the various numerical solutions are comparable: 3.45 $\times 10^{-4}$ for 15 arc-sec, 4.81 $\times 10^{-4}$ for 30 arc-sec and 4.64 $\times 10^{-4}$ for 60 arc-sec. The practical difference between the two algorithms lies in the execution time, which is roughly one order of magnitude faster for GLQ than for adapted FQ (Fig. 3).

## 3 The 2D case

In 2D, the domain $D \subset \mathbf{R^2}$ is discretized into a finite number $N_c^x$ x $N_c^y$ of cells $\{c_{ij}\}$ having constant area $a \times b$, being $a$ the extension along $\hat{x}$, and $b$ the one along $\hat{y}$. The subscripts $i$ and $j$ refer to the nodes in the grid along $\hat{x}$ and $\hat{y}$ respectively. The pair of coordinates $(x,y) \in D$ is a point in the domain. Within each cell $c_{ij}$, the instantaneous, uniform bottom displacement, is again modeled as the difference of two Heaviside functions as (Nosov and Kolesov, 2011):

$$\eta_0^{ij}(x,y) = B_0^{ij}[\theta(x+a) - \theta(x-a)][\theta(y+b) - \theta(y-b)] \tag{8}$$

The sea surface perturbation given by Eq. (16) in Nosov and Kolesov (2011), can be re-stated directly as approximately given by:

$$\xi_0^{ij}(x,y) \simeq U^2 \frac{4B_0^{ij}}{\pi^2} \left( \frac{ab\epsilon}{U^2} + O(\epsilon^4) + \int_{\frac{\epsilon}{U}}^{1} \int_{\frac{\epsilon}{U}}^{1} \frac{cos(Umx)sin(Uma)cos(Uny)sin(Unb)}{mnU^2cosh(kH_0^{ij})} dmdn \right) \tag{9}$$

where $B_0^{ij}$ is the residual bottom deformation and $H_0^{ij}$ is the water depth, taken as positive downward, in the cell $c_{ij}$. The variables of integration $m$ and $n$ represent the spatial wave numbers along $\hat{x}$ and $\hat{y}$, respectively. The variable $k = \sqrt{m^2 + n^2}$
is the modulus of the wave vector. The value of $\epsilon = 10^{-9}$ and $U = \frac{5}{H_0^{ij}}$ are set according to the analysis presented in Section (2.1).

In two dimensions, the LED is defined by a rectangular area surrounding the cell (Fig. 4) in the Cartesian plane:

$$p_{min} = -4H_0^{ij} - max(\frac{a}{2}, \frac{b}{2}) \tag{10}$$

$$p_{max} = 4H_0^{ij} + max(\frac{a}{2}, \frac{b}{2}) \tag{11}$$

$$l_e^x = p_{min}, \; p_{min} + \Delta x, \; p_{min} + 2\Delta x, \ldots, \; p_{max} - \Delta x, \; p_{max} \tag{12}$$

$$l_e^y = p_{min}, \; p_{min} + \Delta y, \; p_{min} + 2\Delta y, \ldots, \; p_{max} - \Delta y, \; p_{max} \tag{13}$$

Numerical solutions for Eq. (9) are computed at each point $(x_p, y_p)$ within Eq. (10), using Gauss-Legendre quadrature with four points. In Section (2.2), we establish that this algorithm is the most efficient for accurately approximating the deformation of the free surface. The numerical scheme for its extension to the 2D case can be found in the Supplementary Materials.

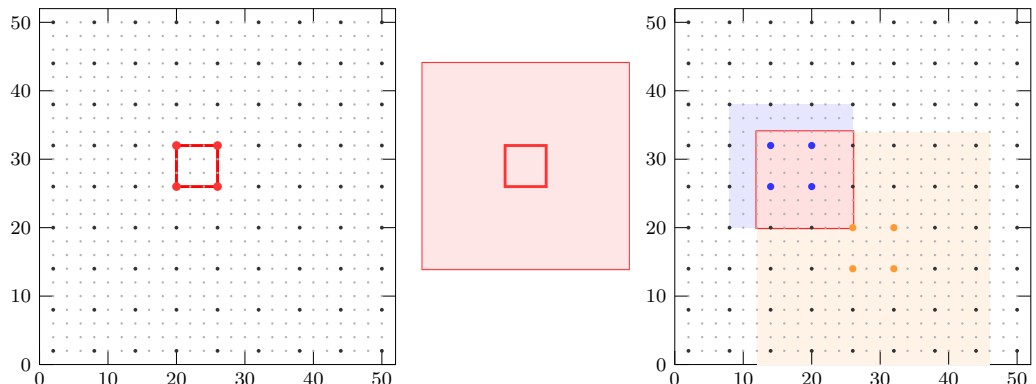

**Figure 4.** *(Left)* A single cell is shown, together with the associated LED. The depth of the water is 3 km in scale. *(Right)* Two cells and their associated LEDs are shown. The sea depths are, to scale, 1.5 km for the blue cell and 3 km for the orange cell. The unit contributions to the total perturbation of the free surface will be superimposed at the intersection of the two LEDs.

## 3.1 Physical Interpretation

Two experiments, detailed in Fig. 5, are conducted in both 1D and 2D. The amplitude of the sea-floor deformation is kept constant at $B_0 = 1$ m in both cases. In the 1D scenario, $H_0$ is initially set to 4 km, with varying cell sizes of 450 m, 900 m, and 1800 m (approximately 15, 30, and 60 arc-sec, respectively). We then explore the case where the Heaviside function (1) encompasses typical wavelengths of coseismic deformation. To establish reasonable orders of magnitude, these wavelengths are set as equivalent to $a = w \cos(\delta)$, representing the projection of a fault width $w$ onto the horizontal plane through the dip angle $\delta$. Specifically, we consider a dip $\delta = 15°$ for a fault plane having width $w = 11$ km and $\delta \in \{20°, 45°\}$ for the one with $w = 27$ km, roughly corresponding to moment magnitudes of $M_w = 6$ and $M_w = 7$, according to the scaling relations presented in Strasser et al. (2010). The initial sea surface height is evaluated through Eq. (5) for each cell length. Figure 5a shows that the smoothing effect increases as the source size decreases, leading to a progressively lower amplitude and narrower width. Sources whose extents are much shorter than the sea depth (15 arc-sec, 30 arc-sec and 60 arc-min) are unable to efficiently uplift the water column. Doubling the source size relative to the sea depth, as in the case of $a \simeq 11$ km, results in an elevation essentially reproducing the unfiltered bottom deformation at the surface, with a maximum of +0.98 m. If the source length is more than four times the local water depth, which is the case of $a \simeq 19$ km and $a \simeq 27$ km, the maximum crest of the water height matches that at the sea-bottom, and filtering affects only the corner of the boxcar. The experiment is replicated for the 2D case, where the values for $a$ are equivalent to those employed in the 1D scenario, and $b$ is set at half of the corresponding $a$ values. In Fig. 5b, a segment of the free-surface disturbance along the $\hat{x}$ axis is depicted, corresponding to the blue line in the top panel of Fig. 5. Simultaneously, Fig. 5c illustrates the profiles acquired along the $\hat{y}$ axis, mirroring the scenario of the magenta plane. The behavior of the model (Eq. 9) aligns with the 1D scenarios for the tested source sizes: a broader extension of the Heaviside function describing coseismic deformation (Eq. 8) results in a less pronounced smoothing effect on the free



surface deformation. Nevertheless, in the 2D case, the maximum free surface elevation values obtained are slightly lower than those in the 1D case: +0.01 m for $a = 450$ m and $b = 225$ m, +0.03 m for $a = 900$ m and $b = 450$ m, +0.1 m for $a = 1800$ m and $b = 900$ m, +0.83 m for $a \simeq 11$ km and $b \simeq 5$ km, +0.97 m for $a \simeq 19$ km and $b \simeq 10$ km and +0.99 m for $a \simeq 26$ km and $b \simeq 13$ km. Figure 5d illustrates the scenario where the 1D unit source length is held constant at $a =\simeq 11$ km as before, with varying depths of 1 km, 4 km and 8 km respectively, corresponding to the average depths of the Mediterranean Sea, the Pacific

Ocean, and trench axes in subduction zones. As the sea depth increases, the sea surface uplift diminishes, accompanied by an expansion in the width of the water height distribution. For $H_0 = 1$ km, the bottom deformation is almost perfectly replicated on the surface in both shape and elevation. With $H_0 = 4$ km, the uplift reaches a maximum of +0.98 m, and the deformation shape is smoothed. When the sea-depth is 8 km, the peak is +0.84 m, and the elevation is redistributed over the tails. A similar trend is observed in Fig. 5e and in Fig.5f, representing two sections of a 2D free-surface perturbation along $\hat{x}$ and $\hat{y}$ axes,

respectively. For $H_0 = 1$ km, results align with the 1D case. The maximum crest is reduced to +0.83 m for $H_0 = 4$ km, and to +0.5 m for $H_0 = 8$ km, indicating that the lateral extension of the coseismic deformation plays a crucial role with varying sea-depths. The findings indicate that the damping level of the 2D filter is closely related to the ratio of wavelengths in the $\hat{x}$ and $\hat{y}$ directions. Specifically, the shorter the deformation is in one direction, the more the smoothing will be pronounced in the other direction. In the Supplementary Materials, we provide a comparison between the scenarios presented in this section and

the outcomes derived from the application of a Kajiura-type filter with different parameterizations of the coseismic deformation and sea-depth values. Additionally, we present the 2D shapes of the free-surface perturbations corresponding to the 1D sections depicted in Fig. 5 (b,c,e,f).

## 3.2    How to construct a local database of unit smoothed initial conditions for tsunami propagation

The mathematical model proposed by Nosov and Kolesov (2011), along with its equivalent scaled version presented in Eq. (9),

is fully characterized by three parameters: the sizes $a$ and $b$ of the rectangular cells by which the domain under study has been discretized and the water depth $H_0^{ij}$ within each cell $c_{ij}$. We note that the amplitude $B_0^{ij}$ of the bottom deformation (Eq. 8) serves in Eq. (9) as a multiplicative constant outside the integral. This observation suggests that Eq. (9) can be independently solved for each $c_{ij} \in D$. Individual solutions can be derived depending solely on the water depth $H_0^{ij}$ inside the cell and the linear dimensions $a, b$ of the cell itself. Without loss of generality, we can set $B_0^{ij} = 1$ within each cell.

The results, each representing a scaled, filtered free surface deformation, can be stored in a repository to be used as a database of unit sources which can be linearly combined to approximate the tsunami initial condition due to any sea bottom deformation (Fig. 4). Assuming sea depth as constant within a cell, Eq. (9) is an analytical solution to the Laplace equation for the scalar potential of fluid velocity. Since the Laplace operator is linear, the superposition principle allows to linearly combine elementary contributions. We designed an algorithm, from now on identified by the acronym LST (Laplacian Smoothing Tool).

A pseudo-code of the LST algorithm, along with its 1D version, is provided in the Supplementary Materials. The LST Bash and Python scripts are also provided (see *Code and data availability*).


**Figure 5.** The problem in two dimensions is illustrated in the upper panel. Considering a rectangle of length $a$ and width $b$, the coseismic deformation $\eta_0$, modelled as described in Eq. (8), is defined with an amplitude $B_0 = 1$ m. This deformation leads to the uplift of the sea surface, causing the perturbation $\xi_0$. Panels (a) and (b) show the 1D perturbations of the free surface, obtained by solving Eq. (5), considering a constant sea depth and a constant cell length, respectively. Panels (b), (c), (e) and (f) show profiles extracted from equivalent 2D cases, evaluated through Eq. (9), along two perpendicular planes (shown in the top panel).




## 4 Test on real events

### 4.1 The tsunamigenic earthquakes in Central Kuril Islands

In the late 2006 and early 2007, two large earthquakes occurred near the Kuril Trench (Fig. 6). Both the events triggered tsunami
waves that spread across the Pacific Ocean and were detected by numerous DART buoys, tide gauges, and bottom pressure
sensors in the far-field. There were no coastal stations in the near-field, but they were located at least 500 km away from the
source (Fujii and Satake, 2008; Rabinovich et al., 2008; Tanioka et al., 2008; Nosov and Kolesov, 2009, 2011). The November
15, 2006 had a a moment magnitude of 8.3 (CMT) and its hypocenter was located at the interface between the subducting
Pacific and the Okhotsk Plates, at coordinates 46.592 °N, 153.266 °E (USGS). A second earthquake followed approximately
two months later, on January 13, 2007. The earthquake was an outer-rise with a normal fault mechanism. The CMT algorithm
estimated a moment magnitude $M_w = 8.1$. The hypocenter was situated along a high-angle fault, beneath the trench slope, at
coordinates 46.243 °N, 154.524 °E (USGS).

We consider here for both events the slip distributions on planar faults published in Lay et al. (2009) (Fig. 6). The slip model
for the 2006 event is based on the inversion of teleseismic P-waves. The slip model for the 2007 event relies on the inversion of
teleseismic P and SH waves. Two different potential fault plane orientations have been identified, one northwest dipping and
the other one southeast dipping. The data did not allow to conclusively determine a preferred plane, leading to the consideration
of both orientations. For the three slip models, we compute the three-dimensional coseismic deformation resulting from each
subfault in which the fault plane is partitioned as a vector $\eta_{\mathbf{0}} = (\eta_{0x}, \eta_{0y}, \eta_{0z})$, where $\eta_{0x}$ and $\eta_{0y}$ denote deformations in
the horizontal directions, modeled as in Eq. (8), while $\eta_{0z}$ represents the vertical component. Subsequently, these individual
contributions are aggregated to form the total sea-floor deformation.

For all the three fault plane geometries, we consider two different models to test the LST algorithm.

The first one is $\eta_{0z}$, the vertical components of the coseismic deformation produced by each subfault (Fig. 6). The second
one, $\eta_{0z} + \eta_{0x} \frac{\partial H_0}{dx} + \eta_{0y} \frac{\partial H_0}{dy}$, accounts for the impact of the horizontal movement of a sloping bottom combined with the vertical
component. The horizontal movement, particularly on steep slopes, such as that of the Kuril Trench, has been identified as a
significant factor in generating seismotectonic tsunamis (Iwasaki, 1982; Tanioka and Satake, 1996; Tanioka and Seno, 2001).
Following the notation in Tanioka and Seno (2001), the latter is identified hereafter as Model A and is equivalent to Eq. (2) in
Tanioka and Satake (1996).

Only for the 2006 megathrust event, we consider also the Model B proposed in Tanioka and Seno (2001), which is a proxy for
the inelastic dislocation of the sediments within the accretionary wedge, due to the movement of the corresponding backstop.
This model is given by $\eta_{0z} + (\eta_{0x} \frac{\partial H_0}{dx} + \eta_{0y} \frac{\partial H_0}{dy}) \frac{h}{w}$, with $h$ and $w$ representing the height of the backstop and the width of the
sediments in the wedge, respectively. For Model B, specific values are chosen, such as $h = 8$ km and $w = 20$ km, which are to
be taken as orders of magnitude derived from the structural and tectonic sections presented in Qiu and Barbot (2022).

A single database of smoothed unit sources, spanning from 152 °E to 157 °E in longitude and from 44 °N to 49 ° N in
latitude, is constructed and encompasses $300 \times 299$ smoothed source elevation values, as detailed in Section (3). For this
application, we use the bathymetry model SRTM30+ (Becker et al., 2009) down-sampled at 1 arc-min.


**2006**  **2007 (NW)**  **2007 (SE)**

**Figure 6.** The fault planes (Lay et al., 2009) and contour lines depicting vertical coseismic deformations (with a 0.25 m interval) for seismic events in the Central Kuril Islands during late 2006 and early 2007 are presented. These deformations are calculated using the Okada (1985) algorithm. In the case of the outer-rise event (early 2007), two distinct slip models are taken into account, as shown in panels (b) and (c). Additionally, the epicenters of the earthquakes, sourced from the USGS catalogue, are marked for reference.

The results for the 2006 event are illustrated in Fig. 7. The sea bottom deformation induced by the vertical component (Fig. 6a) spans from a maximum subsidence of -0.81 m to a maximum uplift of +2.80 m (Fig. 7a).

The output from LST yields an elevation of +2.66 m and a subsidence of -0.77 m (Fig. 7d). The magnitudes are, in modulus, slightly higher than those reported by Nosov and Kolesov in 2011 (+2.55 m upwelling and -0.58 m downwelling) but signifi-
cantly higher than the results obtained by Rabinovich et al. (2008) using a 3D implementation of the Laplace problem for the same case (+1.9 m uplift). These discrepancies in the final water height may be attributed to the different slip and bathymetric models used. The horizontal component substantially displaces the seafloor. In the unfiltered Model A, a peak elevation of +5.64 m and a downwelling of -1.80 m are observed in the deformation field obtained through the traditional approach (Fig. 7b). The application of LST results in a maximum upward movement of +5.37 m and a minimum downfall of -1.70 m (Fig. 7e). The deformation computed through Model B shows a systematically lower maximum crest than in Model A. In projecting




**Figure 7.** Results for the tsunamigenic earthquake occurred on November 15, 2006. The first column depicts the sea-surface distribution arising from a vertical bottom movement. The last two columns present the results obtained with the contribution of the horizontal bottom displacement according to Model A and Model B in Tanioka and Seno (2001). Panel (a) depicts the transect AB where 1D profiles for all the six models have been considered. Panels (g), (h) and (i) show how the simple differences between the unfiltered and filtered initial conditions are spatially distributed.



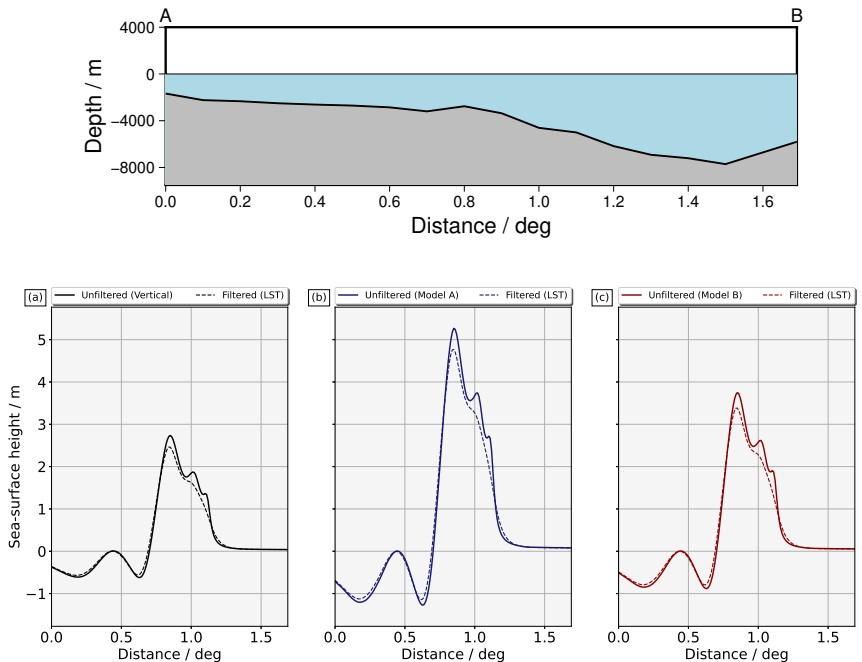

**Figure 8.** The transect AB under consideration is depicted in Fig. 7a. The upper panel illustrates the bathymetric profile. In (a), the profiles are derived from the initial conditions shown in Fig. 7 (a, d, g), taking into account only the vertical component. In (b), the profiles are obtained from the initial conditions in Fig. 7 (d, e, h), incorporating the influence of the horizontal component through Model A. Lastly, in (c), the profiles are extracted from the initial conditions in Fig. 7 (c, f, i), considering the effect of the horizontal component through Model B.

the horizontal deformation onto the vertical plane, the deformation extent in Model B is regulated by the ratio between the backstop height ($h$) and the width of the accretionary wedge ($w$), expressed as $\frac{h}{w}$. Depending on the relative values of the two parameters, particularly when $w$ is significantly higher than $h$, as in this case, this ratio may lead to a damping effect on the contribution from the horizontal component of deformation. The maximum unfiltered uplift for Model B amounts to +3.90

m, lowered to +3.73 m by LST, while the maximum unfiltered depression measures -1.19 m, reduced to -1.12 m when our algorithm is applied (Fig. 7, panels c and f). The last row of Fig. 7) depicts the spatial distributions of differences between the unfiltered and the filtered sea surface height, for all the three considered models. Major differences are concentrated in the proximity of the land, in very shallow waters and towards the Trench side (see Fig. 7 for visualizing the bathymetric changes). For the vertical component $\eta_{0z}$, the maximum differences in uplift and subsidence reach 0.94 m and 0.31 m, respectively (Fig.

7g). For Model A, a significant maximum variation of 1.85 m in elevation and of 0.61 m in depression is observed (Fig. 7h). Similarly, for Model B, the maximum deviation for the positive deformation is 1.30 m and, for the negative one, 0.42 m. The LST appears thus to smooth about three-times more the uplifted sea surface than the subsided one for this event. Figure 8





shows the 1D profiles along the transect AB depicted Fig. 7a for all the nine models. However, it is interesting to note that all the three unfiltered profiles (resulting from the vertical-only coseismic deformation, Model A and Model B) exhibit the three distinct peaks, that are smoothed by the filtering process, resulting in a single pronounced peak.

For the 2007 event, we use only the vertical component (Fig. 6b and Fig. 6c) and the Model A (vertical and projection of the horizontal), as the earthquake occurred in the oceanic crust, relatively far from the sedimentary wedge. The outcomes for the northwest dipping fault plane are depicted in Fig. 9 and in Fig. 10. The sea surface perturbation resulting from the vertical component of the seafloor deformation exhibits a maximum of +0.57 m and a minimum of -5.06 m (Fig. 9a). The application of our LST algorithm yields a positive elevation of +0.29 m and a negative peak of -2.42 m (Fig. 9c), which is less than half the value obtained by translating the seabed deformation to the surface. The filtering effects become more pronounced when considering all three-dimensional components of displacement with Model A, reducing the maximum uplift from +1.13 m to +0.56 m and increasing the maximum depression from -10.15 m to -4.86 m (Fig. 9, panels b and d). The northwest-oriented fault plane, as adopted by Nosov and Kolesov (2011) with a different slip distribution, results in different numerical values, but their application of the Laplace smoothing algorithm's produces almost identical results to those of the LST one, consistently halving the maximum trough. As for the previous case, we show the spatial differences between the unfiltered and the filtered sea surface perturbation in the last row of Fig. 9. Considering vertical-only coseismic deformation, the maximum smoothing in free-surface uplift measures 0.93 m, while that in free-surface subsidence is 2.88 m. When Model A is taken into account, these values are approximately doubled. In particular, the greatest difference observed in uplift is 1.86 m, while in subsidence it is 5.76 m. For this particular event, the smoothing is about three times greater in subsidence than in the uplift and is focused in the proximity of the deepest zones of the Trench (see Fig. 6). Findings for the southeast dipping fault plane are presented in Fig. 11 and in Fig. 12.

When replicating the ocean's bottom deformation caused by the vertical component at sea level (Fig. 6), the negative peak reaches -1.74 m. Through our approach (LST), this value is heightened to -1.44 m. The positive crest is reduced from +0.47 m to +0.31 m (Fig. 11, panels a and c). When the horizontal component is taken into account with Model A, the top height is lessen 0.08 m and the maximum depression 0.32 m (Fig. 11f). The maximum difference for the vertical uplift is 0.83 m, while the one for the vertical subsidence is 0.11 m (Fig. 11e). Minor maximum deviations are observed for Model A: 0.28 m for the positive deformation and 0.46 m for the negative deformation (Fig. 11f). So, for the southeast dipping scenarios, the effect of the filter is quite pronounced on the stronger and shorter-wavelength trough only, which is about two times greater than for the crest. The smoothing effect is more significant for the vertical component, particularly affecting two lobes of deformation positioned on the deep areas of the trench as it can be seen in Fig. (11e) and in Fig. (6c).

## 4.2 Discussion

In Sections (4.1), we investigate events belonging to two major categories of earthquakes, both occurred in the Central Kuril Islands: a megathrust, the 2006 event, and an outer-rise event, represented by the 2007 event. The low-pass filtering effect of the water column appears to be less pronounced for the megathrust, as a result of the flatter dip of the subduction zone with respect to that of the crustal faults considered, which results in longer wavelengths. However, such filtering effect is not negligible,



**Figure 9.** Results for the tsunamigenic earthquake occurred on January 13, 2007. The fault plane is northwest dipping. Panel (a) depicts the transect AB where 1D profiles for all models have been considered. Panels (e) and (f) show the spatial distributions of the simple differences between the unfiltered and filtered initial conditions.

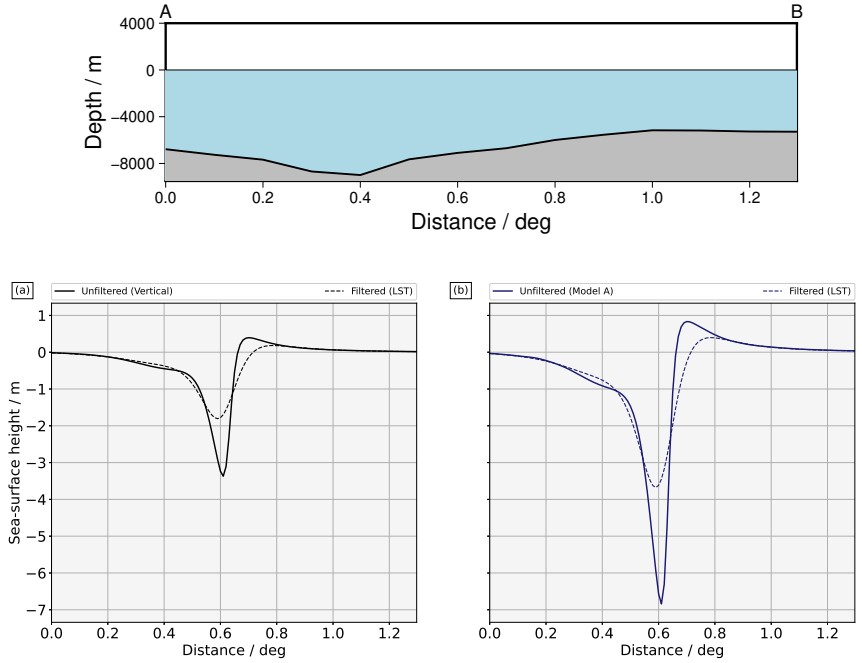

**Figure 10.** The picture refers to the 2007 event in case of a source oriented to the northwest. The transect AB considered is the one depicted in Fig. 9a. The upper panel illustrates the bathymetric profile along it. (a) The profiles are taken from the initial conditions in Figure 9 (a, c, e), considering only the vertical component. (b) The profiles are taken from the initial conditions in Figure 9 (b, d, f), considering the effect of the horizontal component through Model A.

as it can be observed when looking at the Mean Relative Percentage Difference (MRPD) between the LST outputs and the unfiltered free surface deformation for all the nine models. To evaluate the MRPD, the unfiltered free surface deformation $\xi_0^{unf}$ is obtained by copying the coseismic deformation at the free surface, while subtracting the offset due to a positive topographic

340     elevation. In this way, only the perturbation of the water column is considered. The MRPD is then simply computed as:

$$\widehat{e_{MRPD}} = 100 \times mean\left(\left|\frac{\xi_0^{LST} - \xi_0^{unf}}{\xi_0^{unf}}\right|\right) \tag{14}$$

where $\xi_0^{LST}$ is the initial free surface obtained through LST. For the 2006 megathrust in Central Kurils Islands, MRPD is 16.71 % for the vertical component, 21.80 % for Model A and 17.01 % for Model B. The maximum differences between the unfiltered and filtered sea surface height distributions are roughly three times greater in uplift than in subsidence for this

345     earthquake (third row in Fig. 7).

In contrast, the tsunami initial heights are substantially smoothed in the case of the outer-rise 2007 event. For the north-dipping scenarios, the MRPD measured 33.03 % when a vertical-only coseismic deformation is considered, and 46.05 % when



**Figure 11.** Results for the tsunamigenic earthquake occurred on January 13, 2007. The fault plane is southeast dipping. Panel (a) depicts the transect AB where 1D profiles for all models have been considered. Panels (e) and (f) show the spatial distributions of the simple differences between the unfiltered and filtered initial conditions.

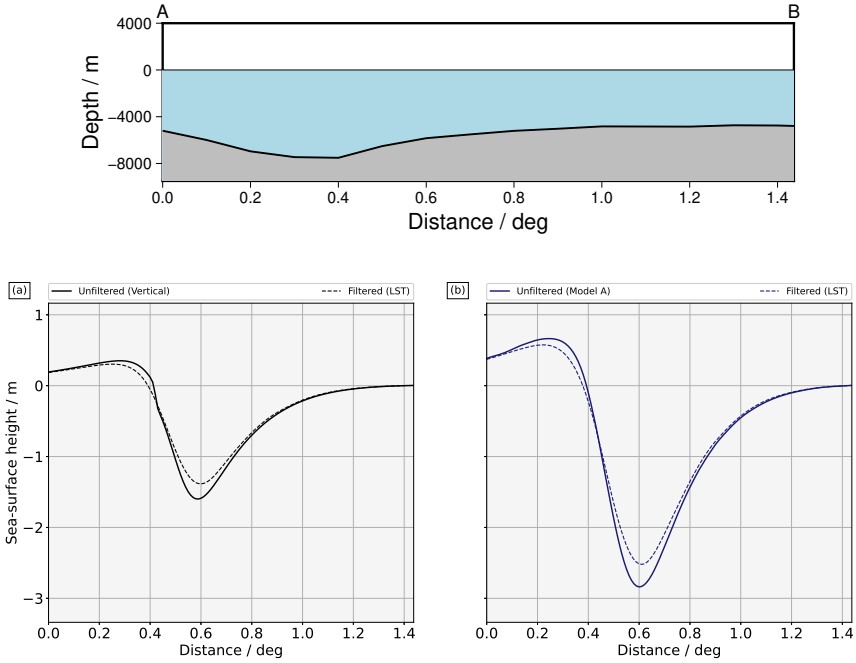

**Figure 12.** The picture refers the 2007 event in case of a source oriented to the southeast. The transect considered is the one depicted in Fig. 11a. Similarly to the Fig. 10, The upper panel illustrates the bathymetric profile along it. (a) The profiles are taken from the initial conditions in Figure 9 (a, c, e), considering only the vertical component. (b) The profiles are taken from the initial conditions in Figure 9 (b, d, f), considering the effect of the horizontal component through Model A.

Model A is taken into account. For the south-east dipping cases, such values are reduced to 16.24 % and 30.92 %, respectively. When considering the maximum spatial differences between the unfiltered and filtered initial conditions, they tend to be roughly three times greater in subsidence than in uplift for all the north-west dipping models. For the south-east cases, such differences are generally greater in subsidence than in uplift, but doubled only for Model A. The areas of coseismic deformations following the megathrust event are in shallower waters if compared to those interested by the outer-rise. The average water depth is ∼2 km - 3 km for the 2006 event, while it amounts to ∼7 km when looking at the 2007 event. Deeper sea depth implies more significantly smoothing of the free surface perturbation. Furthermore, the seafloor deformations associated with the megathrust have much greater length scales than those of the outer-rise (as it can be seen qualitatively in Fig. 6). The same reasoning can be applied to the 2007 event. Despite the similar source area, the two fault planes here considered are different in terms of both direction and value of the dip angle. According to Lay et al. (2009), the southeast-dipping plane exhibits a dip of 59°, while the northwest-dipping plane has a dip of 47°, resulting in different extents of the coseismic deformation. When considering the southeast dipping fault plane, longer wavelengths can be qualitatively observed compared to the opposite dipping model (see Fig. 6b and Fig. 6c). Smaller wavenumbers should be smoothed in this case due to a broader seafloor deformation, contrasting





with the opposite-dipping fault plane where more than half of the deformation is attenuated. We note that for large wavelengths and relatively shallow depths (less than 1 km), there might be no need to account for a smoothing effect on the initial condition (see Fig. 5).

For all the examined events, the horizontal movement of the sloping bottom significantly contributes to the perturbation of the free surface from the equilibrium position. However, we also demonstrate, for example in the case of the 2006 shock, that the initial condition is sensitive to how this horizontal contribution is modeled. In particular, Model A leads to an initial condition where both the maximum uplift and subsidence are more than twice the original unfiltered sea surface deformation. Considering the inelastic component of the coseismic deformation (Model B) would lead to a different outcome, that depend on the size of the accretionary wedge. In general, the LST show a systematic tendency to smooth more the free surface perturbation originated by Model A, for all the scenarios considered. Furthermore, the filtering is more pronounced on the uplift or subsidence, depending on the mechanism of the triggering seismic event.

The LST algorithm is designed for practical applications. Its primary advantage is that it allows the construction of a local database where, depending on the true sea depth, the scaled, smoothed tsunami unit initial conditions are stored to be later used. These unit solutions can be linearly combined, by weighting each of them based on the corresponding coseismic deformation following an event. An example is the database for the Central Kuril Islands, consisting of 89,700 cells. Such database has been created in 2h and 10 min using 6 CPU nodes dual-20-core Intel(R) Xeon(R) Gold 6248 clocked at 2.50 GHz. The execution time required to solve each cell varies with the local sea depth, but it ranges from $\sim 1$ s to $\sim 1$ min, accounting that no inner parallelization is allowed. The linear recombination has been solved in $\sim 9$ min, using a single core Intel(R) Core(TM) i7-10510U CPU clocked at 1.80 GHz. The spatial resolution used is 1 arc-min, and it is noted that higher resolutions might lead to increased computational time. The term "local database" means that the solution depends on the coordinates and local bathymetry of the region. There are plans to distribute it as a service in the future, offering a set of unit solutions based on the corner coordinates of the region of interest. To further enhance efficiency, some proposed ideas include:

1. Since the model's dependence on resolution and water depth is discussed in Section (3), a general database could be constructed considering typical cell dimensions and incremental bathymetric values. This database could then be matched to geographic coordinates by applying latitude correction and binning sea-depth values;

2. The tool could be redesigned to eliminate the need for database construction, potentially parallelizing it to leverage GPU architectures.

## 5 Conclusions

To enhance the computational efficiency and the applicability of the Laplacian Smoothing Algorithm proposed by Nosov and Kolesov (2011), we adopt a strategy informed by numerical analysis. This involves constructing a database of unit initial conditions tailored for tsunami simulations. These sources undergo high-frequency content filtering. Initially addressing the problem in one dimension, we explore the convergence of the integral describing the water height distribution at the sea-surface.





Our findings reveal that only wavenumbers less than $\frac{5}{H}$, with $H$ denoting the flat bathymetry within the cell, are necessary to avoid artifacts when modeling tsunami generation in classic linear potential theory. We conduct a comprehensive comparison
of various numerical quadratures against reference analytic-numeric solutions, evaluating efficiency and accuracy. The model is an analytical solution to the 3D Laplace equation for the fluid velocity potential, which is linear if the sea-bottom does not undergo significant variations within a radius of few wavelengths. Leveraging this linearity and the fact that sea-bottom deformation is linear with respect to the slip, we construct a database of elementary initial conditions. Each entry is scaled by the corresponding bottom displacement.Thus, the methodology allows for considering an arbitrary bottom topography. This
database is then applied to nine different models to obtain the sea surface height distribution following the megathrust and outer-rise events near the Central Kuril Islands in late 2006 and early 2007. We consider the contribution of the vertical component and the impact of horizontal movement of the bottom, highlighting the significance of the latter in earthquakes near steep slopes. Additionally, we demonstrate the sensitivity of the chosen model for representing horizontal components, contingent on the affected area. We observe that the smoothing effect of the water column is particularly evident when considering the horizontal
component, and it is relatively less pronounced in cases of shallow megathrust events, where wavelengths significantly exceed the water depth compared to crustal earthquakes. Despite this, even for interplate earthquakes, the smoothing effect cannot be considered negligible, as it results in approximately a 20 % decrease in the sea-height spatial distribution. We also observe that in general such smoothing effect is more pronounced on the uplifted or subsided free-surface, depending on the mechanism of the seismic event and on its position relative to the coast. In the future, a possible development could involve considering
the case of a time-dependent rupture and assessing its impact on the free-surface deformation. The proposed approach, as well as its applicability to any seafloor displacement and variable bottom topography, may be relevant for practical applications. A further enhancement of its computation performances through HPC architectures could allow the methodology to be used for those studies that require a huge number of simulations, such as long-term probabilistic tsunami hazard assessment (PTHA), and for real-time applications, where the tsunami forecasting needs to be addressed quickly and with the highest possible
accuracy. A further step will be that of studying the sensitivity of the model with respect to different wavelengths and to assess the consequent impact on the inundation.

*Code and data availability.* LST is accessible at the link github.com/abbalice/LST, along with the data used in this study. The subroutine used to perform the Filon quadrature is available at the link https://people.sc.fsu.edu/~jburkardt/m_src/filon/filon.html.

## Appendix A: Acronyms

– LED : Local Extended Domain

– GAQ: Global Adaptive Quadrature

– GLQ: (Adapted) Gauss-Legendre Quadrature



- FQ: (Adapted) Filon Quadrature

- RMSE: Root Mean Squared Error

- LST: Laplacian Smoothing Tool

- MRPD: Mean Relative Percentage Error

*Author contributions.* SL conceived the experiment and JMGV performed some early work on it. AA, MJCD, and JMGV worked on the solution of the integral. AB suggested the rationale for the tool, AA implemented it with contributions from JMGV, MJCD, and SL. AA tested the tool and interpreted the results with contributions from SL, FR, and HBB. AA wrote the manuscript and prepared the figures. All
the authors revised the entire manuscript.

*Competing interests.* The contact author has declared that none of the authors has any competing interests.

*Acknowledgements.* AA acknowledges funding by INGV for her visit at University of Malaga during which part of this study was developed. We thank Maria Lopez Fernandez for fruitful discussions on quadrature formulae and Thorne Lay for providing the slip models for the Central Kuril Islands event. INGV, UMA, and GFZ acknowledge the support by the Horizon EU DT-GEO grant No. 101058129 and Geo-INQUIRE
grant No. 101058518.





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
