# Peer review of "Modeling tsunami initial conditions due to rapid coseismic seafloor displacement: efficient numerical integration and a tool to build unit source databases"

_Natural Hazards and Earth System Sciences, 2024_

## Referee Comment (RC1)

**Referee Report on** "*Modeling tsunami initial conditions due to rapid coseismic seafloor displacement: efficient numerical integration and a tool to build unit source databases*" by Alice Abbate *et al*.

The manuscript under review, authored by **Alice Abbate, José M. González Vida, Manuel J. Castro Díaz, Fabrizio Romano, Hafize Basak Bayraktar, Andrey Babeyko, and Stefano Lorito**, represents a significant contribution to the field of tsunami research. Hailing from esteemed institutions like the Istituto Nazionale di Geofisica e Vulcanologia, the University of Trieste, the University of Malaga, and the GFZ German Research Center for Geosciences, the team brings together a wealth of expertise. Their work focuses on improving the simulation of seismically-induced tsunamis through the numerical evaluation of the Kajiura filter integral for instantaneous vertical seafloor displacements, a topic of paramount importance for both academic research and practical applications in tsunami risk assessment and early warning systems.

The study's strong points lie in its innovative approach to the numerical integration of the Kajiura filter integral and the subsequent development of a tool for constructing tsunami unit source databases. The methodology proposed by the authors to approximate the initial sea level perturbation through a linear combination of elementary sea floor displacements is both efficient and accurate. This allows for rapid simulations of tsunami initiation, which is crucial for improving the timeliness and reliability of tsunami warnings. Moreover, the application of this methodology to the tsunamigenic Kuril earthquake doublet and the consideration of the horizontal contribution to tsunami generation represent notable advancements in the field. The provision of a tool to create tsunami unit source databases offers a valuable resource for the research community and practitioners alike.

However, the manuscript is not without areas that could benefit from further refinement. While the authors have clearly delineated their contributions to the field, the manuscript would benefit from a more comprehensive discussion of the implications of their findings for existing models of tsunami generation and propagation. Specifically, it would be advantageous to elaborate on how their approach compares with current methodologies in terms of computational efficiency, accuracy, and applicability to different seismotectonic settings. Additionally, the paper could be strengthened by addressing potential limitations of the proposed methodology, such as the applicability of their approach to complex bathymetric features and varying sea floor displacements. Furthermore, the manuscript would be enhanced by the inclusion of a more detailed exploration of how the tool for constructing tsunami unit source databases could be integrated into existing tsunami warning systems and risk assessment frameworks.

The manuscript also suffers from a lack of discussion regarding the sensitivity of their model to various parameters, such as the choice of truncation points in the numerical integration and the resolution of the underlying bathymetric data. Addressing these aspects would not

only provide a clearer understanding of the robustness and reliability of their approach but also guide future research efforts in optimizing the model for different scenarios.

The authors assert in the abstract, "We verify that we can satisfactorily approximate the initial sea level perturbation as a linear combination of those induced by the elementary sea floor displacements." While this statement highlights a central aspect of the manuscript's methodology, it is worth noting that this outcome is inherently expected from a theoretical standpoint. This fact naturally follows from the Green's function integral representation of the solution to the Laplace problem for an incompressible and irrotational fluid, combined with the convergence properties of the selected quadrature formula. The linearity of the problem and the superposition principle justify the authors' approach to modelling the initial sea level perturbation. Thus, while the verification of this approach through numerical experiments is valuable for practical applications, the theoretical basis for expecting such a result should not be overlooked.

It is also noted that several relevant references are missing, which could provide a more comprehensive background and context for the study. Incorporating these references would not only enrich the literature review but also position the authors' contributions more clearly within the existing body of knowledge.

Furthermore, the authors mention in the Introduction, "The contribution of the horizontal component to the coseismic deformation can also be important in the presence of steep slopes in the bathymetry (Iwasaki, 1982; Tanioka and Satake, 1996), or in shallow earthquakes resulting in an additional uplift in the accretionary prism (Seno, 2000; Tanioka and Seno, 2001)." This acknowledgment of the significance of horizontal displacements in tsunami generation is crucial. It is pertinent to note that the influence of horizontal seabed movements on tsunami genesis has been previously investigated. For instance, Dutykh et al. (2012) in their study "On the contribution of the horizontal sea-bed displacements into the tsunami generation process" (Ocean Modelling, 56, 43–56, https://doi.org/10.1016/j.ocemod.2012.07.002) offer an early examination of this aspect. Moreover, the application of finite fault solutions to tsunami generation, akin to the methodology employed by Abbate et al., has been discussed in the literature, notably by Dutykh, D., Mitsotakis, D., Gardeil, X., & Dias, F. (2013) in "On the use of the finite fault solution for tsunami generation problems" (Theor. Comput. Fluid Dyn., 27(1–2), 177–199, https://doi.org/10.1007/s00162-011-0252-8). The inclusion of these references could provide a richer historical context to the current study, acknowledging the foundational work upon which the present methodology builds.

In their manuscript, the authors describe different methodologies for modeling the initial conditions of tsunami generation, noting, "Some approaches impose a delta function as the bottom velocity (Levin and Nosov, 2009; Saito, 2017) or transfer to the sea-level the last frame of a time-dependent earthquake rupture simulation (Saito, 2019; Abrahams et al., 2023)." This distinction between methodologies can also be framed within the context of

"passive vs active generation" of tsunamis, as explored in the literature. Specifically, the concept is detailed in Dutykh et al. (https://doi.org/10.1007/978-3-540-71256-5_4), a reference already cited by the authors for other purposes. To enhance the accessibility and comprehensiveness of their discussion, it would be beneficial for the authors to include this terminology, referring to "passive" and "active" tsunami generation. This inclusion would not only align with established nomenclature but also potentially broaden the appeal of their work to readers familiar with these terms from existing literature on tsunami dynamics.

Additionally, from a typographic standpoint, the manuscript could benefit from a refinement in the presentation of mathematical expressions, particularly concerning the notation of (hyperbolic) trigonometric functions. Instead of representing these functions in plain text, such as "*cosh*," the authors should employ the corresponding function notations available in their document preparation system. This practice not only adheres to mathematical typesetting standards but also enhances the clarity and professionalism of the manuscript. Adopting this approach for all mathematical functions within the paper will ensure consistency and improve readability for the audience.

Regarding Figure 5, there is a significant opportunity for improvement in its visual presentation. The current excessively dark style of the figure does not contribute to its clarity or effectiveness in conveying the intended information. To enhance the readability and overall visual appeal of the figure, it is recommended that the authors remove the background grid. This adjustment would simplify the figure's appearance, making it easier for readers to focus on the key data and findings presented. Such a revision would align with best practices in scientific visualization, ensuring that figures serve as effective communication tools within the manuscript.

In conclusion, the manuscript "*Modeling tsunami initial conditions due to rapid coseismic seafloor displacement: efficient numerical integration and a tool to build unit source databases*" by Alice Abbate et al. presents a valuable contribution to the field of tsunami research. However, to fully realise the manuscript's potential as a fantastic scientific article, the authors should consider expanding their discussion of the implications, limitations, and sensitivity of their methodology. Additionally, addressing the issue of missing references will further strengthen the paper. Given these considerations, I recommend a revision of the paper, confident that the authors will address these points constructively, thereby significantly enhancing the value and impact of their work.

---

## Referee Comment (RC4)

[general comments]
The authors of Abbate et al. developed a long-time desired tool to calculate the initial perturbation of the water surface in the tsunami source (Laplacian Smoothing Tool). The LST considers the smoothing effect of the water layer, and therefore significantly improves the accuracy of the input data for numerical tsunami modelling. The linear recombination of the unit sources for the Central Kuril Islands has been solved by in just 9 min, which allows us to hope that the developed tool will be in demand not only in retrospective tsunami studies, but also in real-time tsunami forecast. The paper and its supplementary materials describe the approach underlying LST and the details of the implementation of this approach. The material is well organized, written in clear language, and deserves publication after minor revisions (see comments below).

[specific comments]

The only weakness of the paper is the absence of a detailed comparison of LST with the more accurate methods of initial perturbation calculation. The amplitudes of the Kuril tsunamis calculated using LST are compared with similar amplitudes obtained by Rabinovich et al. 2008 and Nosov & Kolesov 2011. But it is difficult to get any insights from such a comparison of amplitudes alone, especially since the bathymetric and bottom deformation data were different in all these works.... However, considering that the main purpose of the paper was to describe and demonstrate LST, comparison of LST with methods of other authors can be postponed for further research.

The theoretical background of LST is based on Abrahams et al 2023, Davies & Griffin 2018 and Nosov & Kolesov 2011. It is not clear from Section 1 whether any of these papers compared the Kajiura-type filter (with the average ocean depth) and the solution of the full 3D Laplace problem (in the ocean with variable depth). I recommend the authors to emphasize the presence/absence of such a comparison, and to mention the paper by Sementsov & Nosov 2023 (https://doi.org/10.20948/mm-2023-02-06), in which the comparison of the Kajiura filter and the full Laplace problem solution was carried out for a 2D case (0XZ).

In Figure 2, tolerance is shown in colour (without units) and is also plotted on the vertical axis (in %). In the text of the article, the formula for MAE is given first, and the subsequent analysis is carried out in terms of tolerance. I recommend the authors to check the figure once again and briefly describe the connection between MAE and tolerance.

Section 3. The integration limit U and the optimal quadrature method (GLQ) for the 2D case were chosen based on the tests for 1D. A comment may need to be added that the 1D results can indeed be extended to 2D.

145-146: 'It should be recalled that the approximation is valid when both the bathymetry and coseismic displacement vary slowly within such a radius (4H0)'. Are there any quantitative limitations for this 'slowly'? If these limitations are violated, can the result be improved by reducing the cell size? (these questions can be discussed in the Discussion section up to the authors decision).

301-302: 'The LST appears thus to smooth about three-times more the uplifted sea surface than the subsided one for this event'. Why? Probably, because the uplift peak is located in shallow water while the subsidence peak is located in deep water?

305: It is also interesting to note that the filtered and unfiltered peaks are slightly shifted horizontally one relative to the other. Up to the authors decision this fact could be mentioned in the text.

338, 400: "nine models". Why nine, but not seven?
Kuril 2006: Vertical, A, B;
Kuril 2007, northwest dipping: Vertical, A;
Kuril 2007, southeast dipping: Vertical, A.
Seven in total!

[technical corrections]

Figure 5.
In the Figure: panel e is labeled f by mistake.
In the caption (3rd line): replace (b) with (d).
I guess, this Figure can be improved if the authors sign each panel 1D or 2D, respectively, indicate the depth H in panels (a)-(c), and indicate the value of a in panels (d)-(f). The reader can find all this information in the text, but it would be easier to perceive the figure if this information was shown on it directly.

253: typo, remove the 'a'.

298: Replace Fig.7 with Fig.6.

321: The sentence 'Findings...Fig.12' should be moved to the next paragraph.

Revised by Kirill Sementsov

---

## Author Comment (AC1)

The study's strong points lie in its innovative approach to the numerical integration of the Kajiura filter integral and the subsequent development of a tool for constructing tsunami unit source databases. The methodology proposed by the authors to approximate the initial sea level perturbation through a linear combination of elementary sea floor displacements is both efficient and accurate. This allows for rapid simulations of tsunami initiation, which is crucial for improving the timeliness and reliability of tsunami warnings. Moreover, the application of this methodology to the tsunamigenic Kuril earthquake doublet and the consideration of the horizontal contribution to tsunami generation represent notable advancements in the field. The provision of a tool to create tsunami unit source databases offers a valuable resource for the research community and practitioners alike.

Thank you for these encouraging words.

However, the manuscript is not without areas that could benefit from further refinement. While the authors have clearly delineated their contributions to the field, the manuscript would benefit from a more comprehensive discussion of the implications of their findings for existing models of tsunami generation and propagation. Specifically, it would be advantageous to elaborate on how their approach compares with current methodologies in terms of computational efficiency, accuracy, and applicability to different seismotectonic settings. Additionally, the paper could be strengthened by addressing potential limitations of the proposed methodology, such as the applicability of their approach to complex bathymetric features and varying sea floor displacements. Furthermore, the manuscript would be enhanced by the inclusion of a more detailed exploration of how the tool for constructing tsunami unit source databases could be integrated into existing tsunami warning systems and risk assessment frameworks.

The methodology we propose is already applied to a realistic bathymetry and a variable seafloor deformation within the region of interest. Sea depth and seafloor displacement are held constant at all the grid points in a single cell where the integration is evaluated, but vary across adjacent cells.

Nonetheless, as the algorithm itself has not been fully optimized yet, as we extend the domain size and increase the resolution of the bathymetry and the sea floor deformation, some limitations may arise, particularly concerning the computational efficiency. We will deepen the discussion section, enhancing these aspects and, in view of technical enhancements, explaining the applicability of the tool to warning and risk assessment frameworks. Note that here we limited our focus to the generation zone. The propagation and inundation stages will be the subject of a further study.

The manuscript also suffers from a lack of discussion regarding the sensitivity of their model to various parameters, such as the choice of truncation points in the numerical integration and the resolution of the underlying bathymetric data. Addressing these aspects would not only provide a clearer understanding of the robustness and reliability of their approach but also guide future research efforts in optimizing the model for different scenarios.

We will consider which issues need to be further addressed in the relevant section of the manuscript, delving more deeply into what has already been written regarding, for example, the truncation

The authors assert in the abstract, "We verify that we can satisfactorily approximate the initial sea level perturbation as a linear combination of those induced by the elementary sea floor displacements." While this statement highlights a central aspect of the manuscript's methodology, it is worth noting that this outcome is inherently expected from a theoretical standpoint. This fact naturally follows from the Green's function integral representation of the solution to the Laplace problem for an incompressible and irrotational fluid, combined with the convergence properties of the selected quadrature formula. The linearity of the problem and the superposition principle justify the authors' approach to modelling the initial sea level perturbation. Thus, while the verification of this approach through numerical experiments is valuable for practical applications, the theoretical basis for expecting such a result should not be overlooked.

Thank you for this feedback. We reckon that some statements are confusing in the present version. Softening, for example, as suggested, the statement in the abstract will avoid being misleading. The linearity is indeed guaranteed by the already well known theory. However, such linear combinations are addressed via truncated numerical solutions and with a further approximation regarding the effect of the piston-like displacement onto the adjacent cells on a relatively slowly varying bathymetry until the decay distance, following a previously proposed approach. Therefore, we decided to test the accuracy of the method for various model parameterizations, including preservation of linearity, to ensure the final free surface perturbation can be satisfactorily approximated.

It is also noted that several relevant references are missing, which could provide a more comprehensive background and context for the study. Incorporating these references would not only enrich the literature review but also position the authors' contributions more clearly within the existing body of knowledge. Furthermore, the authors mention in the Introduction, "The contribution of the horizontal component to the coseismic deformation can also be important in the presence of steep slopes in the bathymetry (Iwasaki, 1982; Tanioka and Satake, 1996), or in shallow earthquakes resulting in an additional uplift in the accretionary prism (Seno, 2000; Tanioka and Seno, 2001)." This acknowledgment of the significance of horizontal displacements in tsunami generation is crucial. It is pertinent to note that the influence of horizontal seabed movements on tsunami genesis has been previously investigated. For instance, Dutykh et al. (2012) in their study "On the contribution of the horizontal sea-bed displacements into the tsunami generation process" (Ocean Modelling, 56, 43–56, https://doi.org/10.1016/j.ocemod.2012.07.002) offer an early examination of this aspect. Moreover, the application of finite fault solutions to tsunami generation, akin to the methodology employed by Abbate et al., has been discussed in the literature, notably by Dutykh, D., Mitsotakis, D., Gardeil, X., & Dias, F. (2013) in "On the use of the finite fault solution for tsunami generation problems" (Theor. Comput. Fluid Dyn., 27(1–2), 177–199, https://doi.org/10.1007/s00162-011-0252-8). The inclusion of these references could provide a richer historical context to the current study, acknowledging the foundational work upon which the present methodology builds. In their manuscript, the authors describe different methodologies for modeling the initial conditions of tsunami generation, noting, "Some approaches impose a delta function as the bottom velocity (Levin and Nosov, 2009; Saito, 2017) or transfer to the sea-level the last frame of a time-dependent earthquake rupture simulation (Saito, 2019; Abrahams et al., 2023)." This distinction between methodologies

can also be framed within the context of "passive vs active generation" of tsunamis, as explored in the literature. Specifically, the concept is detailed in Dutykh et al. (https://doi.org/10.1007/978-3-540-71256-5_4), a reference already cited by the authors for other purposes. To enhance the accessibility and comprehensiveness of their discussion, it would be beneficial for the authors to include this terminology, referring to "passive" and "active" tsunami generation. This inclusion would not only align with established nomenclature but also potentially broaden the appeal of their work to readers familiar with these terms from existing literature on tsunami dynamics.

Thanks for these suggestions which will help us better positioning within the context of the previous studies on the topic. All these relevant references will be included in the manuscript.

Additionally, from a typographic standpoint, the manuscript could benefit from a refinement in the presentation of mathematical expressions, particularly concerning the notation of (hyperbolic) trigonometric functions. Instead of representing these functions in plain text, such as "cosh," the authors should employ the corresponding function notations available in their document preparation system. This practice not only adheres to mathematical typesetting standards but also enhances the clarity and professionalism of the manuscript. Adopting this approach for all mathematical functions within the paper will ensure consistency and improve readability for the audience.

The mathematical expressions within the manuscript will be modified accordingly for consistency and readability.

Regarding Figure 5, there is a significant opportunity for improvement in its visual presentation. The current excessively dark style of the figure does not contribute to its clarity or effectiveness in conveying the intended information. To enhance the readability and overall visual appeal of the figure, it is recommended that the authors remove the background grid. This adjustment would simplify the figure's appearance, making it easier for readers to focus on the key data and findings presented. Such a revision would align with best practices in scientific visualization, ensuring that figures serve as effective communication tools within the manuscript.

Figure 5 will be changed according to the reviewer's suggestions.

In conclusion, the manuscript "Modeling tsunami initial conditions due to rapid coseismic seafloor displacement: efficient numerical integration and a tool to build unit source databases" by Alice Abbate et al. presents a valuable contribution to the field of tsunami research. However, to fully realise the manuscript's potential as a fantastic scientific article, the authors should consider expanding their discussion of the implications, limitations, and sensitivity of their methodology. Additionally, addressing the issue of missing references will further strengthen the paper. Given these considerations, I recommend a revision of the paper, confident that the authors will address these points constructively, thereby significantly enhancing the value and impact of their work.

Thank you for your time and effort in reviewing the manuscript. Your valuable comments are nothing but helpful in improving the work and will be gladly addressed by the authors.

---

## Author Comment (AC2)

The NHESS manuscript "Modeling tsunami initial conditions due to rapid coseismic seafloor displacement: efficient numerical integration and a tool to build unit source databases" by Abbate et al. develops and describes a computationally efficient procedure to calculate the attenuation of vertical displacement in the water column during tsunami generation.  This is an important study that provides an accurate and efficient method to determine this phenomenon: too often the water column Green's function ("Kajiura filter") is ignored, leading to an overestimation of onshore wave heights, runup, and inundation.   When implemented in the past, it has often been calculated assuming a constant water depth in the source region.  Overall, the study is well conceived and the manuscript is well organized and written.  The detailed description of the algorithm and pseudo-code in the supplement is appreciated for future applications. General suggestions are provided below to revise the text for the NHESS readership as well as specific in-line comments and corrections.  These should all be easily addressed by the authors.

Thank you for taking the time to review our manuscript and for your comments.

General comments:

1.  I very much appreciate the mathematical rigor of the analysis, so often lacking in many geophysical papers (my own included). The Abstract reads well, but some of the introductory text could be made more engaging to a natural hazards and geophysical audience.  For the Introduction, it would be good to describe the objective of the study closer to the top of the section, particularly in terms of implications for tsunami hazard assessment.  For Section 2, I would very much encourage describing the geophysical problem and associated approximations first, before jumping straight into the mathematics.

Thanks for this feedback, which we may use to make the manuscript more readable. We somehow felt that the Introduction needs to be streamlined a bit more. We will strive to improve the Introduction according to the suggestions. We also recognise, thanks for noting it, that a geophysical description of the problem should be provided at the beginning of Section 2. We will modify it accordingly.

2.  The rationale for using box-car source is unclear to me. Is it because of specific analytic/spectral properties?  Alternatively, it would be more harmonious with existing tsunami modeling practice to use vertical seafloor displacements from unit-slip dislocations (i.e., unit fault sources), although granted, this would have to be regional/subduction zone specific.

The box-car source is used to exploit the analytical solution and the subsequent generalization to any discrete input displacement through the linear combination of unitary slip dislocations. We will try to make this clearer in the Introduction and in Section 2. The application of this approach to realistic cases is shown later in

Section 4. The database for a given zone is then constructed by scaling the deformation, i.e. keeping the deformation equal to one in each box-car, without loss of generality.

3. It would be particularly informative to determine the effect on sea-surface elevation profiles of earthquake ruptures that reach to the sea floor and form a scarp. The scarp displacement is obviously attenuated through the water column, but it has been unclear in previous studies what the resulting sea elevation profile is and the effect on the maximum amplitude. Related to this, I'm assuming the 2006 earthquake was not a sea-floor rupturing event but the 2007 earthquake was? It would be helpful to indicate this in the manuscript explicitly.

Thanks for this interesting comment. According to Lay et al. (2009), the 2006 event ruptured very shallowly, but there is no clear evidence that the rupture extended up to the sea floor. The same applies to the 2007 event. We will specify this in the main text, as suggested. We could try to show the scarp displacement with an additional slip model for the 2007 earthquake.

Specific comments:

L64: Which authors are referred to?

Nosov, M. A. and Kolesov, S. V.: Optimal Initial Conditions for Simulation of Seismotectonic Tsunamis, Pure and Applied Geophysics, 168, 1223–1237, https://doi.org/10.1007/s00024-010-0226-6, 2011.

L97: It would be helpful to describe the "Laplacian problem"/equation for the readers here. Referred to later in the manuscript as well.

The Laplacian problem will be inserted in the main text or in the supplementary material.

4: I suspect most readers are familiar with big-O notation, but perhaps not little-o. Helpful to indicate in the Supplement its meaning and how it is derived. Curious that the little-o term is not included in the 2D equation (9) (or supplement eqn. 27).

The term little-o is not included in the 2D equation to simplify the problem. Once the behavior in 1D was known, we decided not to include the error in 2D. However, for the sake of consistency, we will address this issue in an improved version of the manuscript. We will provide some references to better understand the concept of little-o notation.

L120: Because it is used as a reference solution, it would be helpful to know more about the GAQ method, either in the main text or supplement. What properties does it have that makes it more accurate? It would also be helpful to have more description of the Filon quadrature method.

Thanks for this comment. We will provide some more detailed information accordingly.

L122-123: Please indicate specifically how small "u" is related to big "U".

We will address this point.

L148: Please specify how "numerical integration" is performed. (using each quadrature method?)

The adaptive scheme we provide here simply defines the number of wavelength intervals to be included in the integration process. Once this number has been defined, the two quadrature formulae (Gauss-Legendre and Filon) are applied. We will add this explanation in the revised manuscript.

3: It's a little confusing to have the bars in the chart ordered differently than the table directly beneath the chart.

Fig. 3 will be updated according to this comment.

L199-203: Again, it's confusing why an equivalent Heaviside function is used for sea floor displacement rather than directly using the elastic dislocation equations (Okada) with the source parameters as described.

See the comment above. It is because an exact solution exists.

L242-243: It would be helpful indicate the pertinent Laplace equation near the beginning of Section 2.

Thanks for this comment. We will add the Laplace equation at the beginning of Section 2.

---

## Author Comment (AC3)

Since the other reports are already available, I am only providing additional comments.

Thank you for your time in reviewing the manuscript.

Mathematical questions:

1. In equation (4), why is it \epsilon^3?

Looking at the Supplementary Materials, Eq. (1) can be splitted into the sum of three terms, as explained in Eq. (2). To approximate the integral between 0 and $\varepsilon$ , we use the Taylor expansions to the third order of $\frac{sin(ma)}{ma}$ and $cos(mx)$ , respectively . It follows that

$\frac{sin(ma)}{ma} \approx 1 + O(m^2)$ and $cos(mx) \approx 1 + O(m^2)$ . Finally, the integral a

$\int_{0}^{\varepsilon}(1 + O(m^2))(1 + O(m^2))dm$ can be approximated by $\varepsilon + O(\varepsilon^3)$ .

2. In equation (5), the last parenthesis should be after dm.

Thank you for noticing it. This is actually a typo error that will be fixed.

3. I don't understand equation (6).

Equation (6) defines the maximum spatial frequency of the function $cos(mUx_p)sin(mUa)$ , at the numerator of the integrand. The individual frequencies are

given by $\omega_1 = \frac{Ux_p}{2\pi}$ for the cosine, and $\omega_2 = \frac{Ua}{2\pi}$ for the sine. For a given U value,

the maximum spatial frequency is given by $\omega_{max} = Umax(\frac{Ux_p}{2\pi}, \frac{Ua}{2\pi})$ . The idea is to

take a number of points in the integral support to be applied in the quadrature formulae used to approximate the integral in Eq. (5).

4. I don't understand equation (7).

The Nyquist theorem states that a sinusoidal function can be regenerated with no loss of information as long as it is sampled at a frequency greater than or equal to twice per cycle. As the integrand of Eq(5) is a product of sinusoidal functions, in Eq (7) we take the maximum frequency between the one computed in Eq. (5) and a certain $N_s$ value,

which is supposed to be high enough, in order to take the best representation of our integrand for the following quadrature formulae.

5. Figure 3: it is misleading because in the text the authors mention two quadrature formulas and they mention three in the figure. Why is GAQ the groundtruth?

The GAQ here mentioned is the "Global Adaptive Quadrature" ( Shampine, L.: Vectorized adaptive quadrature in MATLAB, Journal of Computational and Applied Mathematics, 211, 131–140, https://doi.org/https://doi.org/10.1016/j.cam.2006.11.021, 2008). This GAQ method

uses adaptive integration points that are very convenient for our case and we used it with a tolerance of $10^{-8}$ . Of course, this method is more expensive (in terms of computational cost) than Gauss-Legendre of Filon methods, but the results can be used as our reference solution. We will specify this is the text.

6. In equation (9), why is it \epsilon^4?

This results from the natural extension to the 2D case of the reasoning applied to the 1D case (question 1).

There are several awkward sentences. Examples are:

1. The last sentence of the abstract

2. The sentence on lines 58/59

Thank you for saying this. We will try to rephrase those sentences in a clearer way.

The last author is missing in the reference Kervella and Dutykh (2007). In the main text, it should read Kervella et al. (2007). Please replace >> and << by their LaTeX notation: \gg and \ll. I would replace the first sentence of Section 2 by: Let R denote the set of real numbers. We consider a domain D \in R. Trigonometric functions inside equations should be written \cosh, \cos, \sin, \max, etc.

Thanks for noticing it. All these points will be addressed in an updated version of the manuscript.

---

## Author Response (AR1)

Dear Editor,

Thank you for your evaluation of our manuscript.
We wish to also thank the reviewers for their valuable comments, which helped us to improve the quality of the manuscript. We have revised the manuscript according to your and their suggestions.
We summarize the main changes with respect to the previous version of the manuscript in what follows.

- We softened or slightly modified some sentences in the Abstract, as recommended by Reviewers 1 and 3;

- We shortened and streamlined the Introduction to make it more to the point and better highlight the study's objective and its implications for hazard assessment, as recommended by both Reviewers 1 and 2. We also incorporated most of the suggested references;

- We added a new section (Section 2) with the scope of writing the Laplacian equation, as suggested by Reviewer 2;

- At the beginning of Section 3 (Section 2 in the original manuscript), we briefly described the problem from a geophysical perspective, as suggested by Reviewer 2;

- In the same section, we clarified the selection of GAQ as the reference solution, addressing the questions of Reviewers 2 and 3, and added a few lines to describe parts of the equations as required by Reviewer 3. We also adjusted the definition of the "tolerance", according to Reviewer 4;

- We removed some sentences from Section 5 (Section 4 in the previous version of the manuscript), to avoid unnecessary details that we think do not contribute to the scope of the manuscript;

- We remark that thanks to the ongoing code optimization effort, the computational time has now been halved compared to the previous version of the manuscript. This is expected to be further reduced in future implementations of the algorithm. We have noted this in the present version of the manuscript;

- In the Supplementary Materials, few more details about the quadrature formulae and the convergence of the integral have been added, according to the comments from Reviewer 2 and 3;

- All the mathematical notations and figures have been modified to meet quality requirements.

Please, find below are our point-by-point replies to the reviewers' comments. The reviewers' comments are marked in black, and our responses are in blue. Line references in our responses correspond to the revised manuscript.

We gratefully acknowledge that the comments helped improve the manuscript.

Best Regards,
Alice Abbate

Reviewer 1

Referee Report on "Modeling tsunami initial conditions due to rapid coseismic seafloor displacement: efficient numerical integration and a tool to build unit source databases" by Alice Abbate et al. The manuscript under review, authored by Alice Abbate, José M. González Vida, Manuel J. Castro Díaz, Fabrizio Romano, Hafize Basak Bayraktar, Andrey Babeyko, and Stefano Lorito, represents a significant contribution to the field of tsunami research. Hailing from esteemed institutions like the Istituto Nazionale di Geofisica e Vulcanologia, the University of Trieste, the University of Malaga, and the GFZ German Research Center for Geosciences, the team brings together a wealth of expertise. Their work focuses on improving the simulation of seismically-induced tsunamis through the numerical evaluation of the Kajiura filter integral for instantaneous vertical seafloor displacements, a topic of paramount importance for both academic research and practical applications in tsunami risk assessment and early warning systems. The study's strong points lie in its innovative approach to the numerical integration of the Kajiura filter integral and the subsequent development of a tool for constructing tsunami unit source databases. The methodology proposed by the authors to approximate the initial sea level perturbation through a linear combination of elementary sea floor displacements is both efficient and accurate. This allows for rapid simulations of tsunami initiation, which is crucial for improving the timeliness and reliability of tsunami warnings. Moreover, the application of this methodology to the tsunamigenic Kuril earthquake doublet and the consideration of the horizontal contribution to tsunami generation represent notable advancements in the field. The provision of a tool to create tsunami unit source databases offers a valuable resource for the research community and practitioners alike.

Thank you for these encouraging words.

However, the manuscript is not without areas that could benefit from further refinement. While the authors have clearly delineated their contributions to the field, the manuscript would benefit from a more comprehensive discussion of the implications of their findings for existing models of tsunami generation and propagation. Specifically, it would be advantageous to elaborate on how their approach compares with current methodologies in terms of computational efficiency, accuracy, and applicability to different seismotectonic

settings. Additionally, the paper could be strengthened by addressing potential limitations of the proposed methodology, such as the applicability of their approach to complex bathymetric features and varying sea floor displacements. Furthermore, the manuscript would be enhanced by the inclusion of a more detailed exploration of how the tool for constructing tsunami unit source databases could be integrated into existing tsunami warning systems and risk assessment frameworks.

Efficiency is not analyzed in the paper. Instead, we made comparisons between different strategies to evaluate the integral. An optimisation effort of the current version of the algorithm is undergoing, but we are planning to further reduce the computational cost in future work, in which a parallel GPU version will be provided. However, in terms of computational cost, we can say that this method is slightly more efficient than a standard Kajiura filter, which makes it applicable to many different tectonic contexts, but the planned GPU implementation is expected to provide a dramatic gain. The methodology we propose has already been applied to realistic bathymetry and variable seabed deformation within the region of interest. Sea depth and seafloor displacement are held constant at all grid points in a single cell where integration is performed, but vary across adjacent cells.
We have improved the Introduction by elaborating on how and why this tool could be integrated into current hazard assessment procedures.
Note that here we limited our focus to the generation zone. The propagation and inundation stages will be the subject of a further study.

The manuscript also suffers from a lack of discussion regarding the sensitivity of their model to various parameters, such as the choice of truncation points in the numerical integration and the resolution of the underlying bathymetric data. Addressing these aspects would not only provide a clearer understanding of the robustness and reliability of their approach but also guide future research efforts in optimizing the model for different scenarios.

There are several parameters involved in the evaluation of Eq. 9 (corresponding to Eq. 4 in the previous version of the manuscript) for which we have actually discussed sensitivity of the results to the choice of their value.
The first one is $\epsilon$, which is only used to estimate the singularity at $m = 0$. For the 1D problem, the error of the approximation depends on $O(\epsilon^3)$, which allows to consider as a small number. Therefore, we set $\epsilon = 10^{-9}$. For this one, we don't see a strong need to analyze it further.
The next parameter is $U$, which represents the upper bound of the support of the integral and establishes the truncation error. According to the analysis in Section 1.1 of the Supplementary Materials, the truncation error depends on $o(\epsilon^{-\frac{UH_0}{2}})$, which suggests that $U$ should be given in terms of $H_0$. In Section 3.1, we considered some test cases for different values of $U$ and $H_0$, concluding that $U = \frac{5}{H_0}$ might be sufficient. Furthermore, the estimation provided for the

truncation is not sharp. If $t \geq 1$, as in our case, then

$$\int_{UH_0}^{\infty} t^{-1}e^{-t}dt \leq \int_{UH_0}^{\infty} e^{-t}dt$$

so that the error should be $o(e^{-UH_0})$. In particular, choosing $U = \frac{5}{H_0}$ gives a truncation error of the order of 0.5 %, which is considered sufficient for practical applications.
Regarding the resolution of the bathymetric data, we carried out some tests in Section 3, (i.e. Section 2 in the original version of the manuscript) regarding the quadrature formulas, to show the impact on both the accuracy and the execution time of different grid resolutions (15, 30 and 60 arcseconds). Regarding the linear combinations to construct the solution for a realistic case, we carried out a rough sensitivity test on the Kuril data set, considering grid resolutions of 2 arc-min, 1 arc-min and 30 arc-sec, finding that there is approximately a "geometrical" factor of 4 for the execution time when halving or doubling the resolution, which we now noted in the revised manuscript at L 410-411.

The authors assert in the abstract, "We verify that we can satisfactorily approximate the initial sea level perturbation as a linear combination of those induced by the elementary sea floor displacements." While this statement highlights a central aspect of the manuscript's methodology, it is worth noting that this outcome is inherently expected from a theoretical standpoint. This fact naturally follows from the Green's function integral representation of the solution to the Laplace problem for an incompressible and irrotational fluid, combined with the convergence properties of the selected quadrature formula. The linearity of the problem and the superposition principle justify the authors' approach to modelling the initial sea level perturbation. Thus, while the verification of this approach through numerical experiments is valuable for practical applications, the theoretical basis for expecting such a result should not be overlooked.

Thank you for this feedback. We reckon that some statements are confusing in the present version. Softening, for example, as suggested, the statement in the abstract will avoid being misleading. The linearity is indeed guaranteed by the already well known theory in case of uniform bathymetry, due to the linearity of the operators. Unfortunately, this is not the case for a varying bathymetry. However, some relevant literature mentioned in the Introduction and Section 2 of the manuscript proved that the linear approximation is accurate enough for practical applications.

It is also noted that several relevant references are missing, which could provide a more comprehensive background and context for the study. Incorporating these references would not only enrich the literature review but also

position the authors' contributions more clearly within the existing body of knowledge.

Furthermore, the authors mention in the Introduction, "The contribution of the horizontal component to the coseismic deformation can also be important in the presence of steep slopes in the bathymetry (Iwasaki, 1982; Tanioka and Satake, 1996), or in shallow earthquakes resulting in an additional uplift in the accretionary prism (Seno, 2000; Tanioka and Seno, 2001)." This acknowledgment of the significance of horizontal displacements in tsunami generation is crucial. It is pertinent to note that the influence of horizontal seabed movements on tsunami genesis has been previously investigated. For instance, Dutykh et al. (2012) in their study "On the contribution of the horizontal sea-bed displacements into the tsunami generation process" (Ocean Modelling, 56, 43–56, https://doi.org/10.1016/j.ocemod.2012.07.002) offer an early examination of this aspect. Moreover, the application of finite fault solutions to tsunami generation, akin to the methodology employed by Abbate et al., has been discussed in the literature, notably by Dutykh, D., Mitsotakis, D., Gardeil, X., & Dias, F. (2013) in "On the use of the finite fault solution for tsunami generation problems" (Theor. Comput. Fluid Dyn., 27(1–2), 177–199, https://doi.org/10.1007/s00162-011-0252-8). The inclusion of these references could provide a richer historical context to the current study, acknowledging the foundational work upon which the present methodology builds.

In their manuscript, the authors describe different methodologies for modeling the initial conditions of tsunami generation, noting, "Some approaches impose a delta function as the bottom velocity (Levin and Nosov, 2009; Saito, 2017) or transfer to the sea-level the last frame of a time-dependent earthquake rupture simulation (Saito, 2019; Abrahams et al., 2023)." This distinction between methodologies can also be framed within the context of "passive vs active generation" of tsunamis, as explored in the literature. Specifically, the concept is detailed in Dutykh et al. (https://doi.org/10.1007/978-3-540-71256-5_4), a reference already cited by the authors for other purposes. To enhance the accessibility and comprehensiveness of their discussion, it would be beneficial for the authors to include this terminology, referring to "passive" and "active" tsunami generation. This inclusion would not only align with established nomenclature but also potentially broaden the appeal of their work to readers familiar with these terms from existing literature on tsunami dynamics.

Thanks for these suggestions which helped us to better position the manuscript within the context of the previous studies on the topic. Almost all these relevant references have been included in the manuscript.

Additionally, from a typographic standpoint, the manuscript could benefit from a refinement in the presentation of mathematical expressions, particularly concerning the notation of (hyperbolic) trigonometric functions. Instead of representing these functions in plain text, such as "cosh," the authors should employ the corresponding function notations available in their document preparation system. This practice not only adheres to mathematical typesetting standards but also enhances the clarity and professionalism of the manuscript. Adopting this approach for all mathematical functions within the paper will ensure consistency and improve readability for the audience.

The mathematical expressions within the manuscript have been modified accordingly for consistency and readability.

Regarding Figure 5, there is a significant opportunity for improvement in its visual presentation. The current excessively dark style of the figure does not contribute to its clarity or effectiveness in conveying the intended information. To enhance the readability and overall visual appeal of the figure, it is recommended that the authors remove the background grid. This adjustment would simplify the figure's appearance, making it easier for readers to focus on the key data and findings presented. Such a revision would align with best practices in scientific visualization, ensuring that figures serve as effective communication tools within the manuscript.

Figure 5 has been changed according to the reviewer's suggestions.

In conclusion, the manuscript "Modeling tsunami initial conditions due to rapid coseismic seafloor displacement: efficient numerical integration and a tool to build unit source databases" by Alice Abbate et al. presents a valuable contribution to the field of tsunami research. However, to fully realise the manuscript's potential as a fantastic scientific article, the authors should consider expanding their discussion of the implications, limitations, and sensitivity of their methodology. Additionally, addressing the issue of missing references will further strengthen the paper. Given these considerations, I recommend a revision of the paper, confident that the authors will address these points constructively, thereby significantly enhancing the value and impact of their work.

Thank you for your time and effort in reviewing the manuscript. Your valuable comments were nothing but helpful in improving the work.

The NHESS manuscript "Modeling tsunami initial conditions due to rapid coseismic seafloor displacement: efficient numerical integration and a tool to build unit source databases" by Abbate et al. develops and describes a computationally efficient procedure to calculate the attenuation of vertical displacement in the water column during tsunami generation. This is an important study that provides an accurate and efficient method to determine this phenomenon: too often the water column Green's function ("Kajiura filter") is ignored, leading to an overestimation of onshore wave heights, runup, and inundation. When implemented in the past, it has often been calculated assuming a constant water depth in the source region. Overall, the study is well conceived and the manuscript is well organized and written. The detailed description of the algorithm and pseudo-code in the supplement is appreciated for future applications. General suggestions are provided below to revise the text for the NHESS readership as well as specific in-line comments and corrections. These should all be easily addressed by the authors.

Thank you for taking the time to review our manuscript and for your comments which helped us to improve the manuscript.

General comments:

- I very much appreciate the mathematical rigor of the analysis, so often lacking in many geophysical papers (my own included). The Abstract reads well, but some of the introductory text could be made more engaging to a natural hazards and geophysical audience. For the Introduction, it would be good to describe the objective of the study closer to the top of the section, particularly in terms of implications for tsunami hazard assessment. For Section 2, I would very much encourage describing the geophysical problem and associated approximations first, before jumping straight into the mathematics.

  Thanks for this feedback. We improved the Introduction according to the suggestions. In particular, we shortened the Introduction, giving more attention to the geophysical problem and elaborating on how and why this tool could be integrated into current hazard assessment procedures, as also suggested by other reviewers (L 44-49). We also recognise, thanks for noting it, that a geophysical description of the problem should have been provided at the beginning of Section 3 (i.e. Section 2 in the previous version of the manuscript). We modified it accordingly (L 87-90).

- The rationale for using box-car source is unclear to me. Is it because of specific analytic/spectral properties? Alternatively, it would be more

harmonious with existing tsunami modeling practice to use vertical
seafloor displacements from unit-slip dislocations (i.e., unit fault sources),
although granted, this would have to be regional/subduction zone spe-
cific.

The box-car source is used to exploit the analytical solution and the
subsequent generalization to any discrete input displacement through
the linear combination of unitary slip dislocations. We made this clearer
in the Introduction (L 44-49) and in L 87-90 of Section 3, which cor-
responds to Section 2 in the previous version of the manuscript. The
application of this approach to realistic cases is shown later in Section
5. The database for a given zone is then constructed by scaling the de-
formation, i.e. keeping the deformation equal to one in each box-car,
without loss of generality.

- It would be particularly informative to determine the effect on sea-surface
  elevation profiles of earthquake ruptures that reach to the sea floor and
  form a scarp. The scarp displacement is obviously attenuated through
  the water column, but it has been unclear in previous studies what the
  resulting sea elevation profile is and the effect on the maximum ampli-
  tude. Related to this, I'm assuming the 2006 earthquake was not a sea-
  floor rupturing event but the 2007 earthquake was? It would be helpful
  to indicate this in the manuscript explicitly.

Thanks for this interesting comment. According to Lay et al. (2009),
the 2006 event ruptured very shallowly, but there is no clear evidence
that the rupture extended up to the seafloor. The same applies to the
2007 event. We specified this in the main text (L 297-299), as suggested.

Specific comments:

L64: Which authors are referred to?

Nosov, M. A. and Kolesov, S. V.: Optimal Initial Conditions for Simulation
of Seismotectonic Tsunamis, Pure and Applied Geophysics, 168, 1223–1237,
https://doi.org/10.1007/s00024-010-0226-6, 2011. Lines 59-71 of the previous
version of the manuscript have been summarized in L 34-39 of the present ver-
sion.

L97: It would be helpful to describe the "Laplacian problem"/equation for the
readers here. Referred to later in the manuscript as well.

We inserted the Laplacian problem as a new section (Section 2) in the main text.

4: I suspect most readers are familiar with big-O notation, but perhaps not little-o. Helpful to indicate in the Supplement its meaning and how it is derived. Curious that the little-o term is not included in the 2D equation (9) (or supplement eqn. 27).

We included a brief explanation in the Supplementary Materials, at L 21-22 and at L 49.

L120: Because it is used as a reference solution, it would be helpful to know more about the GAQ method, either in the main text or supplement. What properties does it have that makes it more accurate? It would also be helpful to have more description of the Filon quadrature method.

Thanks for this comment. We provided some more details about GAQ in L 134-140 of the revised manuscript, and we gave more description of the Filon quadrature at L 96-100 of the Supplementary Materials.

L122-123: Please indicate specifically how small "u" is related to big "U".

We addressed this point, clarifying the notation of Section 3.1 (which corresponds to Section 2.1 in the previous version of the manuscript).

L148: Please specify how "numerical integration" is performed. (using each quadrature method?)

The adaptive scheme we provide here simply defines the number of wavelength intervals to be included in the integration process. Once this number has been defined, the two quadrature formulae (Gauss-Legendre and Filon) are applied. We added this explanation in the revised manuscript. To make the concept clearer, we have moved L 146-150 of Section 3.2 (i.e. Section 2.2 in the earlier version of the manuscript) to L 128-133 of Section 3.1 (i.e. Section 2.1 in the earlier version of the manuscript), and added L 141-143 and L 161-165.

3: It's a little confusing to have the bars in the chart ordered differently than the table directly beneath the chart.

Fig. 3 has been updated according to this comment.

L199-203: Again, it's confusing why an equivalent Heaviside function is used

for sea floor displacement rather than directly using the elastic dislocation equations (Okada) with the source parameters as described.

See the comment above. It is because an exact solution exists.

L242-243: It would be helpful indicate the pertinent Laplace equation near the beginning of Section 2.

Thanks for this comment. We added a new section (Section 2) to describe the Laplace equation.

Since the other reports are already available, I am only providing additional comments.

Thank you for your time in reviewing the manuscript.

Mathematical questions:

1. In equation (4), why is it $\epsilon^3$?

   Looking at Section 1.1 of the Supplementary Materials, Eq. (1) can be splitted into the sum of three terms, as explained in Eq. (2). To approximate the integral between 0 and $\epsilon$, we use the Taylor expansions to the third order of $\frac{sin(ma)}{m}$ and $\cos(mx)$, respectively . It follows that $\frac{sin(ma)}{m} \simeq 1 + O(m^2)$ and $\cos(mx) \simeq 1 + O(m^2)$. Finally, the integral $\int_0^\epsilon (1 + O(m^2))(1 + O(m^2))dm = \epsilon + O(\epsilon^3)$. In the revised version of the manuscript, Eq. (4) became Eq. (9), and this explanation was added in Section 1.1 of the Supplementary Materials, L 15-21.

2. In equation (5), the last parenthesis should be after dm.

   Thank you for noticing it. This is actually a typo error that has been fixed. In the revised version of the manuscript, Eq. (5) became Eq. (9).

3. I don't understand equation (6).

   Equation (11) (which corresponds to Eq. (6) in the previous version of the manuscript) defines the maximum spatial frequency of the function

$\cos(mUx_p)\sin(mUa)$, at the numerator of the integrand. The individual frequencies are given by $w_1 = \frac{Ux_p}{2\pi}$ for the cosine, and $w_2 = \frac{Ua}{2\pi}$ for the sine. For a given $U$ value, the maximum spatial frequency is given by $w_{max} = U\max(w_1, w_2)$. The idea is to take a number of points in the integral support to be applied in the quadrature formulae used to approximate the integral in Eq. (10) (which corresponds to Eq. (5) in the previous version of the manuscript). This explanation has been incorporated in the revised manuscript at L 173-179.

4. I don't understand equation (7).

   The equation comes from the Nyquist theorem. As the integrand of Eq. (10) (i.e. Eq. (5) in the previous version of the manuscript) is a product of sinusoidal functions, in Eq. (12) (i.e. Eq.(7) in the previous version of the manuscript) we take the maximum frequency between the one computed in Eq. (11) and a certain $N_s$ value, which is supposed to be high enough, in order to take the best representation of our integrand for the following quadrature formulae. This explanation has been integrated into what was already written in the main text at L 180-186.

5. Figure 3: it is misleading because in the text the authors mention two quadrature formulas and they mention three in the figure. Why is GAQ the groundtruth?

   The GAQ here mentioned is the "Global Adaptive Quadrature" ( Shampine, L.: Vectorized adaptive quadrature in MATLAB, Journal of Computational and Applied Mathematics, 211, 131–140, 10.1016/j.cam.2006.11.021, 2008). This GAQ method uses adaptive integration points that are very convenient for our case and we used it with a tolerance of $10^{-8}$. Of course, this method is more expensive (in terms of computational cost) than Gauss-Legendre of Filon methods, but the results can be used as our reference solution. We specified this is the text (L 134-140).

6. In equation (9), why is it $\epsilon^4$?

   This results from the natural extension to the 2D case of the reasoning applied to the 1D case (question 1).

There are several awkward sentences. Examples are:

1. The last sentence of the abstract

2. The sentence on lines 58/59

Part of the introduction has been summarized and the sentence in question has been deleted. We rephrased the last sentence of the abstract, thanks.

The last author is missing in the reference Kervella and Dutykh (2007). In the main text, it should read Kervella et al. (2007). Please replace ¿¿ and ¡¡ by their LaTeX notation: $\gg$ and $\ll$. I would replace the first sentence of Section 2 by: Let R denote the set of real numbers. We consider a domain $D \in R$. Trigonometric functions inside equations should be written $\cosh, \cos, \sin, \max,$ etc.

Thanks for noticing it. All these points were addressed in the updated version of the manuscript.

Reviewer 4
[general comments]

The authors of Abbate et al. developed a long-time desired tool to calculate the initial perturbation of the water surface in the tsunami source (Laplacian Smoothing Tool). The LST considers the smoothing effect of the water layer, and therefore significantly improves the accuracy of the input data for numerical tsunami modelling. The linear recombination of the unit sources for the Central Kuril Islands has been solved by in just 9 min, which allows us to hope that the developed tool will be in demand not only in retrospective tsunami studies, but also in real-time tsunami forecast. The paper and its supplementary materials describe the approach underlying LST and the details of the implementation of this approach. The material is well organized, written in clear language, and deserves publication after minor revisions (see comments below).

Thank you for your time in reviewing this manuscript and for your consideration.

[specific comments]

The only weakness of the paper is the absence of a detailed comparison of LST with the more accurate methods of initial perturbation calculation. The amplitudes of the Kuril tsunamis calculated using LST are compared with similar amplitudes obtained by Rabinovich et al. 2008 and Nosov & Kolesov 2011. But it is difficult to get any insights from such a comparison of amplitudes alone, especially since the bathymetric and bottom deformation data were different in all these works.... However, considering that the main purpose of the paper was to describe and demonstrate LST, comparison of LST with methods of other authors can be postponed for further research.

Thank you very much for this comment. As you said, the purpose of this study was to provide an alternative methodology to accurately and efficiently approximate the initial condition for tsunami propagation, and we decided to provide only a benchmark 'qualitatively' to leave room for analyzing the impact of different choices (e.g. different slip models or different parameterisations for horizontal components) on the assessment of the initial sea surface perturbation.

The theoretical background of LST is based on Abrahams et al 2023, Davies & Griffin 2018 and Nosov & Kolesov 2011. It is not clear from Section 1 whether any of these papers compared the Kajiura-type filter (with the average ocean depth) and the solution of the full 3D Laplace problem (in the ocean with variable depth). I recommend the authors to emphasize the presence/absence of such a comparison, and to mention the paper by Sementsov & Nosov 2023 (https://doi.org/10.20948/mm-2023-02-06), in which the comparison of the Kajiura filter and the full Laplace problem solution was carried out for a 2D case (0XZ).

Thank you for this hint, we have addressed this point and included this reference.

In Figure 2, tolerance is shown in colour (without units) and is also plotted on the vertical axis (in %). In the text of the article, the formula for MAE is given first, and the subsequent analysis is carried out in terms of tolerance. I recommend the authors to check the figure once again and briefly describe the connection between MAE and tolerance.

Thank you for noticing it. We modified Figure 2 according to the proposed suggestions.

Section 3. The integration limit U and the optimal quadrature method (GLQ) for the 2D case were chosen based on the tests for 1D. A comment may need to be added that the 1D results can indeed be extended to 2D.

We better specified this in the revised version of the manuscript (L 208-210).

145-146: 'It should be recalled that the approximation is valid when both the bathymetry and coseismic displacement vary slowly within such a radius (4H0)'. Are there any quantitative limitations for this 'slowly'? If these limitations are violated, can the result be improved by reducing the cell size? (these questions can be discussed in the Discussion section up to the authors decision).

Thank you for this question. We have relied on what is specified in the relevant literature, although we have not carried out any further tests to quantify the possible limitations of such approximations. However, it is in our plans to include this type of analysis in future work. That sentence has been deleted in the revised version of the manuscript, to avoid confusion.

301-302: 'The LST appears thus to smooth about three-times more the uplifted sea surface than the subsided one for this event'. Why? Probably, because the uplift peak is located in shallow water while the subsidence peak is located in deep water?

Thank you for this question. It is the other way around: the uplift peak is located in deep water, while the subsidence peak is in shallow water (Fig. 6). Sorry for the confusion. We revised the text accordingly.

305: It is also interesting to note that the filtered and unfiltered peaks are slightly shifted horizontally one relative to the other. Up to the authors decision this fact could be mentioned in the text.

Thank you for this hint, we decided to not comment on this in the revised version, as we found no clear interpretation to mention for it beyond a natural effect of the filter.

338, 400: "nine models". Why nine, but not seven? Kuril 2006: Vertical, A, B; Kuril 2007, northwest dipping: Vertical, A; Kuril 2007, southeast dipping: Vertical, A. Seven in total!

Thank you for noticing it. This is actually a typo error that has been fixed. They were nine in a previous version of the manuscript indeed. We have eliminated some because they were not adding too much to the presentation.

[technical corrections] Figure 5. In the Figure: panel e is labeled f by mistake. In the caption (3rd line): replace (b) with (d). I guess, this Figure can be improved if the authors sign each panel 1D or 2D, respectively, indicate the depth H in panels (a)-(c), and indicate the value of a in panels (d)-(f). The reader can find all this information in the text, but it would be easier to perceive the figure if this information was shown on it directly.

253: typo, remove the 'a'.

298: Replace Fig.7 with Fig.6.

321: The sentence 'Findings...Fig.12' should be moved to the next paragraph.

All these points have been addressed in the revised manuscript.

Revised by Kirill Sementsov